# Centriolar satellites assemble centrosomal microcephaly proteins to recruit CDK2 and promote centriole duplication

Andrew Kodani[1], Timothy W Yu[2†], Jeffrey R Johnson[3†], Divya Jayaraman[2], Tasha L Johnson[3], Lihadh Al-Gazali[4], Lāszló Sztriha[4], Jennifer N Partlow[2], Hanjun Kim[1], Alexis L Krup[1], Alexander Dammermann[5], Nevan J Krogan[3], Christopher A Walsh[2], Jeremy F Reiter[1*]

[1]Department of Biochemistry and Biophysics, Cardiovascular Research Institute, University of California, San Francisco, San Francisco, United States; [2]Howard Hughes Medical Institute, Boston Children's Hospital, Boston, United States; [3]Department of Cellular and Molecular Pharmacology, University of California, San Francisco, San Francisco, United States; [4]Department of Paediatrics, College of Medicine and Health Sciences, United Arab Emirates University, Al-Ain, United Arab Emirates; [5]Max F. Perutz Laboratories, University of Vienna, Vienna, Austria

*For correspondence: Jeremy.
Reiter@ucsf.edu

†These authors contributed
equally to this work

Competing interests: The
authors declare that no
competing interests exist.

Reviewing editor: W James
Nelson, Stanford University,
United States

**Abstract** Primary microcephaly (MCPH) associated proteins CDK5RAP2, CEP152, WDR62 and CEP63 colocalize at the centrosome. We found that they interact to promote centriole duplication and form a hierarchy in which each is required to localize another to the centrosome, with CDK5RAP2 at the apex, and CEP152, WDR62 and CEP63 at sequentially lower positions. MCPH proteins interact with distinct centriolar satellite proteins; CDK5RAP2 interacts with SPAG5 and CEP72, CEP152 with CEP131, WDR62 with MOONRAKER, and CEP63 with CEP90 and CCDC14. These satellite proteins localize their cognate MCPH interactors to centrosomes and also promote centriole duplication. Consistent with a role for satellites in microcephaly, homozygous mutations in one satellite gene, *CEP90*, may cause MCPH. The satellite proteins, with the exception of CCDC14, and MCPH proteins promote centriole duplication by recruiting CDK2 to the centrosome. Thus, centriolar satellites build a MCPH complex critical for human neurodevelopment that promotes CDK2 centrosomal localization and centriole duplication.

## Introduction

At the heart of the centrosome, the principle microtubule-organizing center of a mammalian cell, are two centrioles. During S phase, centrioles are duplicated through the coordinated assembly of centriolar complexes near the proximal ends of the preexisting centrioles. Centriole duplication is coupled to the cell cycle preventing under- or over-duplication, either of which can disrupt spindle formation and chromosomal segregation (*Hinchcliffe et al., 1999*; *Lacey et al., 1999*; *Meraldi et al., 1999*; *Vidwans et al., 1999*; *Haase et al., 2001*).

Within centrosomes, a pair of centrioles are surrounded by pericentriolar material and centriolar satellites, 70–100 nm diameter electron dense structures that participate in microtubule-based transport (*Kubo et al., 1999*; *Dammermann and Merdes, 2002*; *Kodani et al., 2010*). Centriolar satellites contain PCM1, thought to function in the delivery of proteins to the centrosome (*Kim et al., 2008*; *Kodani et al., 2010*). Consistent with this hypothesis, depletion of PCM1 reduces the centrosomal localization of select proteins (*Dammermann and Merdes, 2002*; *Kim et al., 2004*, *2008*;

**eLife digest** When a cell divides, the chromosomes that contain the genetic blueprint for the cell must be replicated and shared between the two new cells. A structure called the centrosome organizes the cellular machinery that separates the chromosome copies during cell division. At the center of each centrosome are two cylindrical microtubule-based structures called centrioles.

Mutations in certain proteins that interact with the centrosome cause a neurodevelopmental disorder called primary microcephaly. People born with microcephaly have unusually small heads and brains. As a result, they may have difficulties with mental tasks. Scientists do not know exactly how these 'microcephaly-associated' proteins normally interact with the centrosomes or what they do at the centrosomes, so it is difficult to work out what goes wrong in people with microcephaly. One idea is that the proteins help to duplicate the centrioles before a cell divides. If this duplication does not occur, a cell cannot divide properly; so, people with mutations that interfere with centriole duplication cannot grow enough brain cells.

Now, Kodani et al. have examined how these microcephaly-associated proteins work with 'satellite' proteins that congregate near the centrosome to duplicate centrioles. The satellite proteins help to recruit four microcephaly-associated proteins to the centrosome, where they are built into a ring. The microcephaly-associated proteins congregate at the centrosome in a particular order, with each protein recruiting the next one in the sequence. Once all four are in place near the centrosome, an enzyme that helps to duplicate the centrioles joins them.

Further experiments suggest that mutations that affect one of the satellite proteins—known as CEP90—may cause microcephaly. Future analysis of how microcephaly-associated genes work may reveal the cell biological mechanisms by which centrioles participate in brain development.

---

*Kodani et al., 2010*). Centriolar satellite proteins also promote ciliogenesis and centriole duplication through only partially elucidated mechanisms (*Kim et al., 2004*; *Sedjai et al., 2010*; *Kim and Rhee, 2011*; *Kim et al., 2012*; *Stowe et al., 2012*; *Firat-Karalar et al., 2014*; *Klinger et al., 2014*).

In addition to chromosomal missegregation, altered centriole biogenesis is associated with human developmental growth disorders, such as primordial dwarfism and MCPH (*Mochida and Walsh, 2001*, *2004*; *Woods et al., 2005*; *Griffith et al., 2008*; *Rauch et al., 2008*; *Thornton and Woods, 2009*). Many of the genes mutated in MCPH encode centrosomal proteins (*Bond et al., 2005*; *Zhong et al., 2005*; *Kumar et al., 2009*; *Guernsey et al., 2010*; *Nicholas et al., 2010*; *Yu et al., 2010*; *Sir et al., 2011*; *Lin et al., 2013*). The association of MCPH proteins with the centrosome is evolutionarily conserved with *Caenorhabditis elegans* (*Delattre et al., 2006*; *Strnad and Gonczy, 2008*). Three MCPH proteins, CEP152, CEP135 and STIL, interact with and promote the centrosomal localization of SAS4 (also known as CPAP or CENPJ) (*Strnad and Gonczy, 2008*; *Cizmecioglu et al., 2010*; *Dzhindzhev et al., 2010*; *Sir et al., 2011*; *Brown et al., 2013*; *Lin et al., 2013*). Failure to recruit SAS4 can attenuate centriole elongation and duplication (*Schmidt et al., 2009*; *Comartin et al., 2013*; *Lin et al., 2013*). Together, these observations have raised the possibility that CEP152, CEP135 and STIL promote the recruitment of proteins to the centrosome to facilitate centriole duplication. However, how these MCPH-associated proteins localize to the centrosome and how they promote centriole duplication have remained largely elusive.

Apart from CEP152, CEP135, STIL and SAS4, the protein products of other MCPH-associated genes, including WDR62, CDK5RAP2 and CEP63, participate in centriole biogenesis and function (*Barrera et al., 2010*; *Nicholas et al., 2010*; *Yu et al., 2010*; *Sir et al., 2011*). Whether and, if so, how these proteins function together are unclear. We tested the hypothesis that these MCPH-associated proteins biochemically interact and cooperate in a shared mechanism of centriole biogenesis. To test this hypothesis, we identified interactors of each MCPH-associated protein and found that the MCPH proteins CDK5RAP2, CEP152, WDR62 and CEP63 physically associate with each other. Moreover, they form a hierarchy in which each is required to localize another to the centrosome, and that this stepwise assembly at the centrosome is essential to promote centriole duplication.

In addition to interacting with each other, the MCPH-associated proteins CDK5RAP2, CEP152, WDR62 and CEP63 each interacts with a cognate centriolar satellite proteins. Their associated centriolar satellite partners are required for the localization of the interacting MCPH-associated

protein to the centrosome. Consistent with a role in building the MCPH protein complex at the centrosome, centriolar satellites, like their MCPH-associated proteins, are necessary for centriole duplication to occur efficiently. Thus, paralleling the hierarchy of MCPH-associated proteins, there is a hierarchy of satellite proteins, each of which participates in the centriolar localization of an MCPH-associated protein. We found that a homozygous, missense mutation affecting one of these centriolar satellite components, *CEP90*, is associated with microcephaly, further validating the functional connection between centriolar satellites and the function of the previously defined MCPH-associated proteins.

How do the centrosomal MCPH-associated proteins promote centriole duplication? We found that both the MCPH-associated proteins and their centriolar satellite partner proteins are required for the centrosomal localization of CDK2, a cyclin-dependent kinase with established roles in both cell cycle progression and centriole duplication (*Hinchcliffe et al., 1999*; *Meraldi et al., 1999*). Thus, MCPH-associated proteins and centriolar satellites cooperate to ensure the stepwise recruitment to the centrosome of additional MCPH-assocated proteins, culminating in bringing CDK2 to the centrosome and duplicating the centrioles.

## Results

### MCPH-associated proteins interact and hierarchically localize to the centrosome

To determine which centrosomal proteins CDK5RAP2, CEP152, WDR62 and CEP63 associate with, we immunoprecipitated endogenous CDK5RAP2, CEP152, WDR62, and CEP63 from HeLa cells and identified co-precipitating proteins by LC-MS/MS. c-Myc, a non-centrosomal protein, served as a negative control in our immunoprecipitation experiments. While we detected a number of interacting proteins, we focused our study on proteins previously reported to localize to the centrosome and implicated in centriole duplication. Specifically, we used the data from Jakobsen et al. and Balestra et al. to identify likely centrosomal proteins and those required for centriole biogenesis, respectively (*Jakobsen et al., 2011*; *Balestra et al., 2013*). The mass spectrometry confirmed previously identified interactions, such as the association of CEP152 with CDK5RAP2 and CEP63 (*Sir et al., 2011*; *Brown et al., 2013*; *Lukinavicius et al., 2013*; *Firat-Karalar et al., 2014*), and suggested that CDK5RAP2, CEP152, WDR62 and CEP63 may interact with each other (*Supplementary file 1*). Consistent with this possibility, co-immunoprecipitation confirmed that CDK5RAP2, CEP152, WDR62 and CEP63 interact with each other, but not with an unrelated centriole component, CP110 (*Chen et al., 2002, Figure 1A*).

To confirm the centrosomal enrichment of MCPH-associated proteins, we fractionated centrosomes from HeLa cells by sucrose gradient centrifugation. CDK5RAP2, CEP152, WDR62 and CEP63 co-fractionated with the centrosomal protein CP110 (*Figure 1B*). Taken together, these findings suggest that CDK5RAP2, CEP152, WDR62, and CEP63 interact with each other at the centrosome.

As MCPH-associated proteins localize to the centrosome, we investigated whether the interaction of CDK5RAP2, CEP152, WDR62 and CEP63 reflects a shared centrosomal function. Consistent with previous studies, depletion of *CEP152* or *CEP63* disrupted centriole duplication (*Figure 1—figure supplement 1C,G*; *Brown et al., 2013*). In these, and other similar experiments described below, we observed a mixture of cells in S phase (as determined by nuclear Cyclin A localization) with two and three centrioles, instead of four centrioles as observed in scrambled control siRNA (SC)-transfected cells: the images presented in the figures are of cells in which three centrioles are observed. Similar to the depletion of *CEP152* and *CEP63*, knockdown of *CDK5RAP2* or *WDR62*, inhibited centriole duplication (*Figure 1—figure supplement 1A,E*).

In contrast, mouse embryonic fibroblasts (MEFs) expressing a fusion between a truncated Cdk5rap2 and βGEO or decreased levels of Wdr62 exhibit centriole overaccumulation (*Barrera et al., 2010*; *Chen et al., 2014*), suggesting that efficient depletion of CDK5RAP2 and WDR62 has different consequences than truncations or partial reduction (*Figure 1—figure supplement 1B,F*). To confirm that CDK5RAP2 (also called *CEP215*) is required for centriole duplication, we repeated *CDK5RAP2* knockdown using two previously published siRNAs (*Graser et al., 2007*). Both of these *CDK5RAP2* siRNAs also reduced the percentage of S phase cells with two pairs of centrioles (*Figure 1—figure supplement 2A,B*). Instead of four centrioles, the majority of S phase *CDK5RAP2*, *CEP152*, *WDR62* or

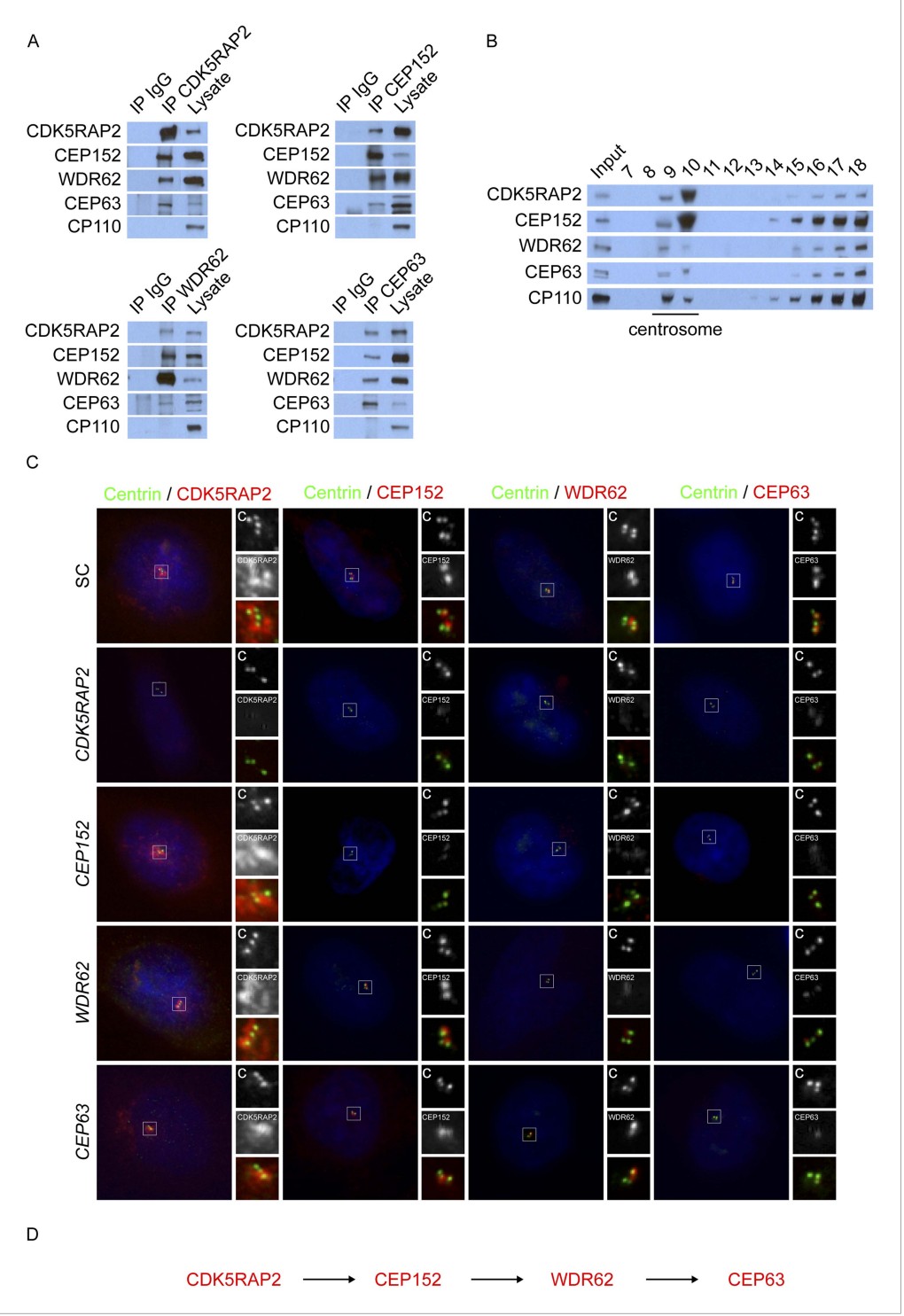

**Figure 1**. CDK5RAP2, CEP152, WDR62, and CEP63 interact at the centrosome and localize to centrioles in a stepwise manner. (**A**) We immunoprecipitated endogenous CDK5RAP2, CEP152, WDR62, and CEP63 from HeLa total cell lysates. Precipitation and co-precipitation were detected using antibodies specific to CDK5RAP2, CEP152, WDR62, and CEP63. The centriole component CP110 served as a negative control. (**B**) Sucrose gradient fractions of HeLa cell lysates were analyzed by immunoblot with antibodies to CP110, CDK5RAP2, CEP152, WDR62, and CEP63.
(**C**) Immunofluorescence of S phase scrambled control (SC), *CDK5RAP2, CEP152, WDR62,* and *CEP63* siRNA-treated HeLa cells co-stained for Centrin ('c', green) to visualize centrioles, CDK5RAP2 (red), CEP152 (red), WDR62 (red), and CEP63 (red), and nuclei (DAPI, blue). The inset shows magnified images of the boxed region. (**D**) Our findings indicate
*Figure 1. continued on next page*

*Figure 1. Continued*

that CDK5RAP2, recruits CEP152 to the centrosome, which in turn recruits WDR62 and CEP63. Scale bars indicate 5 μm for all images.

The following figure supplements are available for figure 1:

**Figure supplement 1**. CDK5RAP2, CEP152, WDR62 and CEP63 are required for centriole duplication.

**Figure supplement 2**. CDK5RAP2/CEP215 promotes centriole duplication and centrosome organization.

**Figure supplement 3**. Centriole duplication defects in *CDK5RAP2, CEP152, WDR62* and *CEP63*-depleted mitotic cells.

**Figure supplement 4**. CDK5RAP2, CEP152, WDR62 and CEP63 form a ring near parental centrioles.

**Figure supplement 5**. CDK5RAP2, CEP152, WDR62 and CEP63 do not control the stability of their binding partners.

**Figure supplement 6**. CDK5RAP2, CEP152, WDR62 and CEP63 localize in a hierarchical manner at the centrosome.

**Figure supplement 7**. CEP63-depletion using a previous published siRNA destabilizes MCPH proteins.

*CEP63*-depleted cells possessed only two or three centrioles (*Figure 1C*, depicting instances of three centrioles). A reduced number of centrioles in S phase could be caused by either a failure or delay in centriole duplication. Therefore, we also examined the number of centrioles in mitotic SC, *CDK5RAP2, CEP152, WDR62* and *CEP63*-depleted cells (*Figure 1—figure supplement 3A*). Depleting MCPH-associated proteins also caused a significant decrease in the number of centrioles in mitosis, suggesting that these MCPH-associated proteins have a shared function in promoting centriole duplication.

As depletion of each of these four MCPH-associated proteins, CDK5RAP2, CEP152, WDR62 and CEP63, has similar effects on centriole duplication, we examined whether they co-localize. We discovered that, like CDK5RAP2, CEP152, and CEP63, WDR62 localizes to the interphase centrosome (*Figure 1C*). To more precisely identify where WDR62 localizes at the centrosome, we used structured illumination microscopy. Also like CDK5RAP2, CEP152, and CEP63, WDR62 is present in a toroidal structure with a diameter of 400–500 nm adjacent to SAS6, a component of the proximal end of procentrioles (*Figure 1—figure supplement 4A*, *Strnad et al., 2007*; *Sir et al., 2011*). We found that the appearance of CDK5RAP2 differed depending on whether it was imaged using SIM or diffraction limited microscopy, which may be attributable to the limitations of SIM in imaging low contrast samples such as the CDK5RAP2 outside of the toroidal structure.

Given their indistinguishable sub-centrosomal localization, we hypothesized that these MCPH proteins participate in each other's localization. Surprisingly, we found that there is a hierarchy by which they localize to the centrosome. CEP152, WDR62, and CEP63 failed to localize to the centrosome in the absence of *CDK5RAP2* (*Figure 1C*, and quantitated in *Figure 1—figure supplements 2E, 7A*). As this localization dependency differed from that reported by Firat-Karalar et al. we confirmed that the centrosomal localization of CEP152 depends on CDK5RAP2 using four non-overlapping siRNAs (*Figure 1C* and *Figure 1—figure supplement 2C*).

Conversely, depletion of *CEP152, WDR62* or *CEP63* had no effect on CDK5RAP2 localization, indicating that CDK5RAP2 has a unique role in localizing other MCPH-associated protein to centrosomes (*Figure 1C*, and quantitated in *Figure 1—figure supplement 6A*). The stability of CEP152, WDR62 and CEP63 was unaltered in *CDK5RAP2*-depleted cells, indicating that their mislocalization was not due to degradation (*Figure 1—figure supplements 3D, 6A*).

CEP152 has previously been shown to be essential for the centrosomal localization of CEP63 (*Brown et al., 2013*; *Lukinavicius et al., 2013*). We confirmed this finding and extended it by finding that depletion of *CEP152* also prevented WDR62 from localizing to the centrosome (*Figure 1C*, and quantitated in *Figure 1—figure supplement 6A*). Consistent with a hierarchy of MCPH-assocated

proteins, only CEP63 failed to localize to the centrosome in the absence of *WDR62*, while CDK5RAP2, CEP152 and WDR62 all properly localized to centrosomes in the absence of *CEP63* (*Figure 1C*, and quantitated in *Figure 1—figure supplement 6A*). Similar to the depletion of *CDK5RAP2,* the protein levels of all four proteins were unaltered in these MCPH protein knockdown cells (*Figure 1—figure supplement 5A*). Thus, CEP152, WDR62 and CEP63 have progressively more restricted roles in MCPH protein localization to centrosomes than does CDK5RAP2 (*Figure 1D*).

Previous studies have reported that CEP152 and CEP63 depend on each other to localize to the centrosome (*Sir et al., 2011*; *Brown et al., 2013*; *Lukinavicius et al., 2013*). In support of these findings, we confirmed that CEP63 failed to localize to the centrosome in *CEP152*-depleted cells with no detectable changes in protein stability (*Figure 1C* and *Figure 1—figure supplement 5A*). Contrary to previous findings, we did not detect a decrease of centrosomal CEP152 in *CEP63*-depleted cells (*Figure 1C* and *Figure 1—figure supplement 6A*). A *CEP63* siRNA described by Brown et al. (*CEP63 NB*) efficiently depleted CEP63 and disrupted centriole duplication (*Figure 1—figure supplement 7A,B*). Surprisingly, *CEP63 NB* also reduced the centrosomal localization of CDK5RAP2, CEP152 and WDR62 (*Figure 1—figure supplement 7C,D*). *CEP63 NB* also reduced the protein levels of CDK5RAP2, CEP152 and WDR62 (*Figure 1—figure supplement 7E,F*). Whereas *CEP63 NB* destabilized CEP152, the other *CEP63* siRNAs used in this study did not, suggesting that depletion of CEP63 may not necessarily lead to CEP152 destabilization and reduced centrosomal CEP152.

Taken together, these results suggest that the centrosomal localization of CEP152, WDR62 and CEP63 depends on CDK5RAP2, localization of WDR62 and CEP63 depends on CEP152, and localization of CEP63 depends on WDR62, indicating a hierarchical localization scheme. As loss of any of these MCPH proteins disrupts centriole duplication or stability, we propose that the ordered accumulation of MCPH proteins at the centrosome is critical for centriole duplication or stability.

## MCPH proteins interact with centriolar satellite components

In addition to MCPH-associated proteins, CDK5RAP2, CEP152, WDR62 and CEP63 copurified with other proteins (*Supplementary file 1*). Some of these co-purifying proteins, including CEP72, CEP131, CEP90, KIAA0753 and CCDC14, localize to centriolar satellites (*Oshimori et al., 2009*; *Jakobsen et al., 2011*; *Kim and Rhee, 2011*; *Hall et al., 2013*; *Firat-Karalar et al., 2014*). MCPH copurifying proteins also included SPAG5 (also called ASTRIN), a mitotic spindle component, which has not been previously reported to localize to interphase centrosomes (*Gruber et al., 2002*; *Thein et al., 2007*). CEP72, SPAG5 CEP131, CEP90, and CCDC14 all colocalize with PCM1 at centriolar satellites throughout the cell cycle (*Figure 2A*).

As KIAA0753 similarly localized to satellites, we refer to it as MOONRAKER (MNR), a satellite from the Ian Fleming novel of the same name (*Figure 2A*). In contrast to PCM1 and other centriolar satellite proteins, SPAG5 also localizes to the microtubules of the mitotic spindle. The spindle localization of SPAG5 may reflect its ability to directly bind microtubules and organize the spindle during mitosis (*Gruber et al., 2002*; *Fitzgerald et al., 2006*; *Yuan et al., 2009*; *Schmidt et al., 2010*).

As CEP72, SPAG5, CEP131, MNR, CEP90 and CCDC14 localize to centriolar satellites throughout the cell cycle, similar to PCM1 (*Figure 2A*), we used co-immunoprecipitation of endogenous proteins to investigate whether they interact with the core centriolar satellite component PCM1. We confirmed previous reports that CEP72, CEP131, MNR, CEP90 and CCDC14 associate with PCM1 (*Figure 2B*, *Kim and Rhee, 2011*; *Staples et al., 2012*; *Stowe et al., 2012*; *Hall et al., 2013*; *Firat-Karalar et al., 2014*). In addition, we identified SPAG5 as a PCM1 interacting protein (*Figure 2B*). Reciprocal immunoprecipitations of these new satellite proteins confirmed that they interact with PCM1, but not with another unrelated centrosomal protein, CP110 (*Figure 2C–H*). The colocalization with PCM1 and the physical association with PCM1 indicate that SPAG5, like CEP72, CEP131, MNR, CEP90 and CCDC14, is a centriolar satellite component.

Previous studies have shown that centriolar satellites bring cargo proteins to centrosomes in a way that depends on intact microtubules (*Dammermann and Merdes, 2002*; *Kodani et al., 2010*). After brief treatment with nocodazole to depolymerize microtubules, PCM1 forms large cytoplasmic aggregates together with satellite cargo proteins (*Dammermann and Merdes, 2002*). Like satellite cargo proteins, microtubule depolymerization caused a portion of CDK5RAP2, CEP152, WDR62 and CEP63 to relocalize to PCM1 aggregates (*Figure 2—figure supplement 1A,B*). In contrast, γ-tubulin

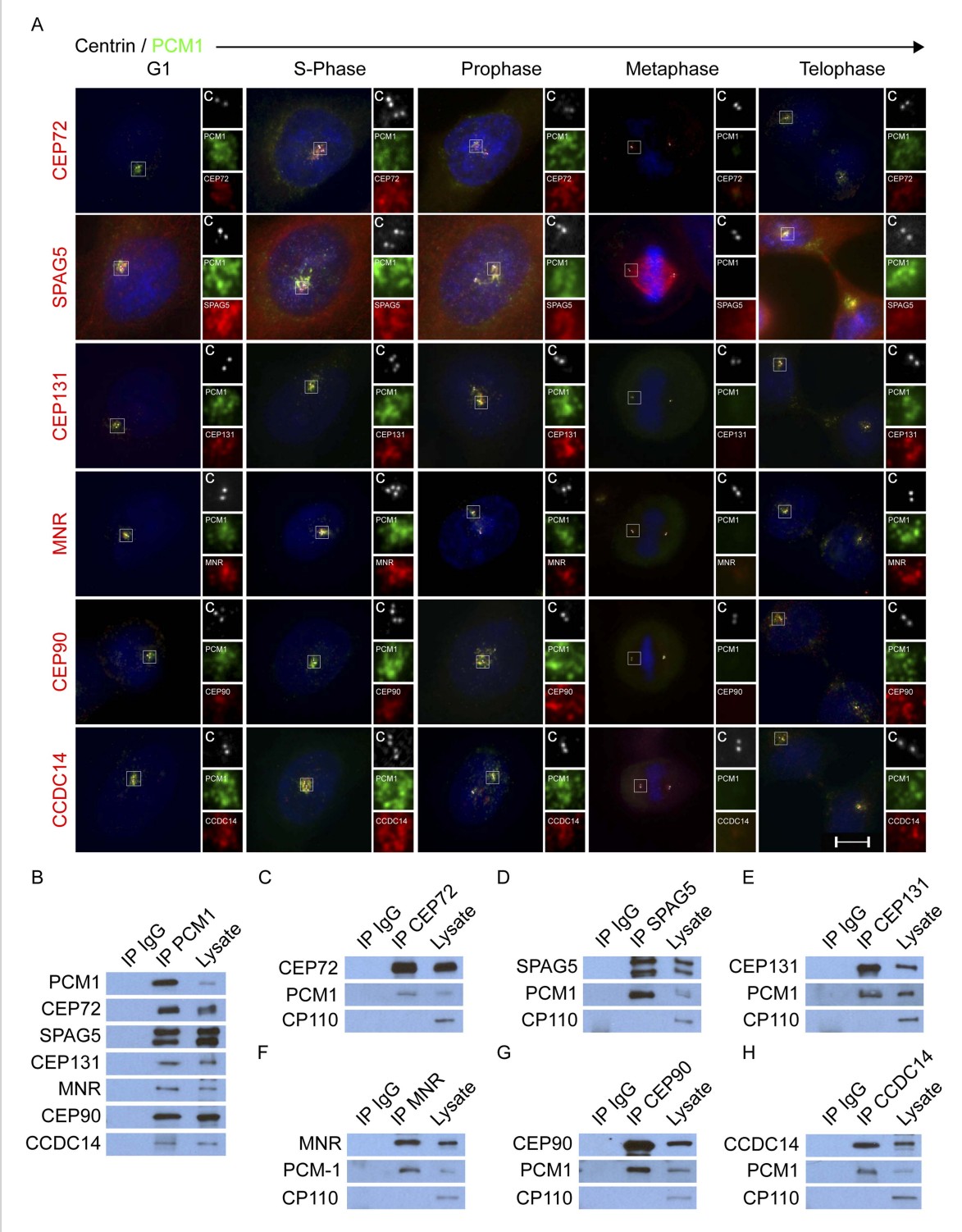

**Figure 2**. CEP72, SPAG5, CEP131, MNR, CEP90 and CCDC14 are centriolar satellite components. (**A**) Immunofluorescence microscopy of asynchronously growing HeLa cells co-stained with antibodies to Centrin ('c', white), PCM1 (green), CEP72 (red), SPAG5 (red), CEP131 (red), MNR (red), CEP90 (red), CCDC14 (red). Cell cycle stage was indicated by the number of Centrin ('c', green) foci and DNA condensation (Hoechst, blue). The insets show magnified images of the boxed regions. Scale bars indicate 5 µm for all images. (**B**) HeLa total cell lysates were subjected to immunoprecipitation of PCM1 and c-Myc (IgG), which served as a negative control throughout. Precipitating proteins were subjected to immunoblotting for PCM1, CEP72, SPAG5, CEP131, MNR, CEP90, and CCDC14. (**C–H**) Reciprocal immunoprecptation of PCM1 and copurified proteins. Endogenous proteins were immunoprecipitated with antibodies to CEP72, SPAG5, CEP131, MNR, CEP90, CCDC14, and a negative control IgG. Complexes were immunoblotted with antibodies to PCM1,

*Figure 2. continued on next page*

*Figure 2. Continued*

CEP72, SPAG5, CEP131, MNR, CEP90, and CCDC14. Scale bars indicate 5 μm for all images.

The following figure supplement is available for figure 2:

**Figure supplement 1**. CDK5RAP2, CEP152, WDR62 and CEP63, but not γ-tubulin, localize to the centrosome in a microtubule-dependent manner.

did not accumulate at the nocodazole-induced PCM1 aggregates, suggesting that not all centrosomal proteins move from centriolar satellites to the centrosome in a microtubule-dependent manner (*Figure 2—figure supplement 1A,B*). The accumulation of four MCPH-associated proteins in PCM1 aggregates upon nocodazole treatment raised the possibility that they are cargoes that centriolar satellites help transport to the centrosome. Tests of this hypothesis are described in the following four sections.

## Centriolar satellite proteins CEP72 and SPAG5 are required for MCPH-associated protein CDK5RAP2 to localize to the centrosome

Mass spectrometry of CDK5RAP2 coprecipitating proteins identified the centriolar satellite proteins, CEP72 and SPAG5 (*Supplementary file 1*). We confirmed that the endogenous CDK5RAP2 protein interacts with CEP72 and SPAG5 by reciprocal immunoprecipitation (*Figure 3A,B*). As CDK5RAP2 is part of a complex with other MCPH-associated proteins, we examined whether CEP72 and SPAG5 also interacted with CEP152, WDR62 and CEP63. Interestingly, immunoprecipitation of CEP72 and SPAG5 did not co-precipitate CEP152, WDR62 or CEP63, suggesting that CEP72 and SPAG5 interact with CDK5RAP2, but not the other MCPH-associated proteins (*Figure 3—figure supplement 1A*). In addition to interacting with CDK5RAP2, we found that CEP72 and SPAG5 immunoprecipitated each other (*Figure 3C*).

To begin to test our hypothesis that centriolar satellite components help MCPH-associated proteins localize to the centrosome, we assessed whether CEP72 and SPAG5 were required to localize CDK5RAP2 to the centrosome. In HeLa cells depleted of either *CEP72* or *SPAG5*, there was a sharp decrease in the localization of CDK5RAP2 at the centrosome (*Figure 3D*). We also found that the protein levels of CDK5RAP2 were reduced in *CEP72* and *SPAG5*-depleted cells (*Figure 3E*). Interestingly, in addition to destabilizing CDK5RAP2, the stability of CEP72 and SPAG5 was reduced in *SPAG5* and *CEP72*-depleted cells, respectively (*Figure 3E*). As in HeLa cells, CEP72 and SPAG5 were essential for stabilizing CDK5RAP2 and localizing it to the centrosome in U2OS cells (*Figure 3—figure supplement 2A–C*). These findings suggest that the centriolar satellite proteins CEP72 and SPAG5 interact with CDK5RAP2, stabilize each other and CDK5RAP2, and promote the localization of CDK5RAP2 to the centrosome.

Previous studies have shown that CEP72 and SPAG5 are each required for centrosome maturation during formation of the mitotic spindle (*Oshimori et al., 2009*; *Schmidt et al., 2010*). Given that CEP72 and SPAG5 are required for CDK5RAP2 centrosomal localization, we hypothesized that, during interphase, CEP72 and SPAG5 may be important for centriole biogenesis. Therefore, we examined whether depletion of either *CEP72* or *SPAG5* phenocopies the effects of *CDK5RAP2* depletion on centriole duplication. Similar to the loss of CDK5RAP2, greater than 78% of *CEP72*-depleted and 83% of *SPAG5*-depleted S phase cells had fewer than four centrioles, indicating that these proteins may function together with CDK5RAP2 to promote centriole duplication or stability (*Figure 3F–I*).

Because CEP72 and SPAG5 are required for the centrosomal localization of CDK5RAP2, and CDK5RAP2 is required for the centrosomal localization of other MCPH-associated proteins CEP152, WDR62 and CEP63, we examined whether the loss of SPAG5 or CEP72 altered the localization of the CDK5RAP2-dependent MCPH proteins. Similar to the depletion of *CDK5RAP2*, depletion of *SPAG5* or *CEP72* caused CEP152, WDR62 and CEP63 localization to become significantly decreased at S phase centrosomes of both HeLa and U2OS cells (*Figure 3—figure supplements 2C, 3A–C*). Thus, the centriolar satellite components SPAG5 and CEP72 are required for the centrosomal localization of CDK5RAP2 and its dependent proteins and the promotion of centriole duplication (*Figure 3J*).

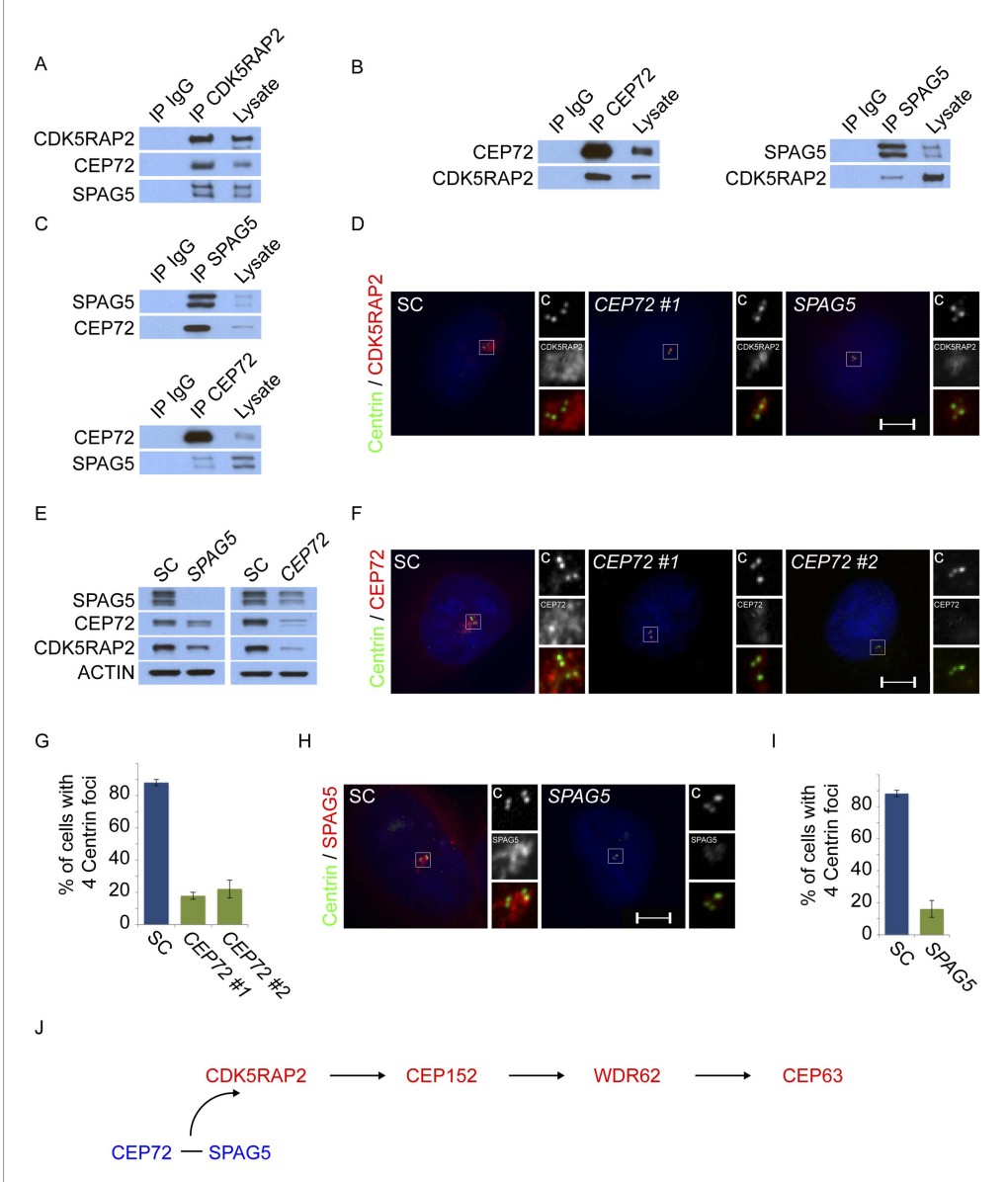

**Figure 3.** Centriolar satellite components CEP72 and SPAG5 stabilize and localize CDK5RAP2 to the centrosome to promote centriole duplication. (**A**) HeLa total cell lysates were subjected to immunoprecipitation of CEP72, SPAG5, and a negative control. Precipitating proteins were subjected to immunoblotting for CEP72, SPAG5 and CDK5RAP2. (**B**) The interactions between CEP72 and SPAG5 with CDK5RAP2 were validated by immunoprecipitation using an antibody to CDK5RAP2. Endogenous CDK5RAP2 was immunoprecipitated and complexes were immunoblotted with antibodies to CEP72, SPAG5 and CDK5RAP2. (**C**) Immunoprecipitated CEP72, SPAG5, and a negative control were immunoblotted using antibodies to CEP72, SPAG5 and a negative control. (**D**) HeLa cells in S phase transfected with SC, *CEP72 #1* or *SPAG5* siRNA were co-stained for CDK5RAP2 (red) and Centrin ('c', green) to visualize centrioles. (**E**) Total cell lysates of SC, *CEP72 #1*, and *SPAG5* siRNA transfected HeLa cells were analyzed by immunoblotting with antibodies to CEP72, SPAG5, and CDK5RAP2. Actin served as a loading control. (**F**) SC, *CEP72 #1*, *CEP72 #2* siRNA transfected HeLa cells were co-stained with CEP72 (red) and Centrin ('c', green) to visualize centrioles. (**G**) Percentage of S phase cells with four Centrin foci in SC, *CEP72 #1*, *CEP72 #2* siRNA treated HeLa cells. Error bars represent ±s.d. throughout. (**H**) S phase HeLa cells transfected with SC or *SPAG5* siRNA co-stained with Centrin ('c', green) and SPAG5 (red). (**I**) Quantification of S phase SC and *SPAG5*-depleted HeLa cells with four Centrin foci. (**J**) Our findings suggest that CEP72 and SPAG5 are required to localize CDK5RAP2 to the centrosome, which itself recruits CEP152, WDR62 and CEP63. Centrosome localized MCPH proteins are represented in red and centriolar satellites in blue. For all quantifications at least 100 cells were counted per experiment (n = 3), p < 0.005 (paired t-test) for SC vs *CEP72 #1, #2* and

*Figure 3. continued on next page*

*Figure 3. Continued*

*SPAG5* siRNA transfected cells. Scale bars indicate 5 µm for all images.

The following figure supplements are available for figure 3:

**Figure supplement 1**. CEP72 and SPAG5 interact with CDK5RAP2 but not CEP152, WDR62 or CEP63.

**Figure supplement 2**. CEP72, SPAG5 and CDK5RAP2 promote centriole duplication and centrosome organization in U2OS cells.

**Figure supplement 3**. CEP72 and SPAG5 are required to localize CDK5RAP2, CEP152, WDR62 and CEP63 to the centrosome.

## Centriolar satellite protein CEP131 is required for MCPH-associated protein CEP152 to localize to the centrosome

Mass spectrometry of CEP152 coprecipitating proteins suggested that CEP152 interacts with CEP131, a centriolar satellite protein involved in genomic stress responses and ciliogenesis (*Supplementary file 1B*, *Staples et al., 2012*; *Hall et al., 2013*; *Villumsen et al., 2013*). We confirmed that CEP152 and CEP131 interact by reciprocally co-immunprecipitating the endogenous proteins (*Figure 4A*). We hypothesized that the relationship between CEP152 and its satellite interactor, CEP131, might parallel the relationship we had identifed for CDK5RAP2 and its centriolar satellite interactors, CEP72 and SPAG5. Therefore, we examined whether CEP131 was required for the centrosomal localization of CEP152. Interestingly, CEP152 was greatly reduced at the centrioles in *CEP131*-depleted cells (*Figure 4B* and *Figure 4—figure supplement 1A*). Unlike the relationship between CDK5RAP2, CEP72 and SPAG5, The loss of CEP152 from the centrosome of *CEP131*-depleted cells was not due to destabilization, as CEP152 protein levels were unchanged by *CEP131* knockdown (*Figure 4C*). Similar to the siRNA-mediated depletion, we observed disrupted centrosomal localization of CEP152 in mouse embryonic fibroblasts (MEFs) derived from embryonic day 14.5 *Cep131^{gt/gt}* embryos (*Figure 4—figure supplement 2A*, *Hall et al., 2013*).

To investigate whether the failure of CEP152 to localize to centrosomes in *CEP131*-depleted cells compromises the interaction of CEP152 with its interacting MCPH-associated proteins, we immunoprecipitated endogenous CEP152 from control and *CEP131*-depleted cells and examined whether CDK5RAP2 and WDR62 co-precipitated. Although depletion of *CEP131* had no effect on CDK5RAP2, CEP152 or WDR62 stability, CEP131 was required for CEP152 to co-precipitate either CDK5RAP2 or WDR62 (*Figure 4—figure supplement 3*). The finding that the centriolar satellite-mediated centrosomal localization of CEP152 is critical for its interaction with its MCPH-associated partner proteins is consistent with the idea that the MCPH-associated proteins interact specifically at the centrosome.

As CEP152 is required for centriole duplication, we examined whether its centriolar satellite interactor, CEP131, is also required for centriole duplication. Depletion of *CEP131* in HeLa cells and examination of Centrin and CEP131 revealed that greater than 84% of *CEP131*-depleted cells in S phase failed to properly duplicate their centrioles (*Figure 4D,F*). Early passage *Cep131^{gt/gt}* MEFs displayed similar defects in centriole duplication (*Figure 4—figure supplement 2B,C*).

As depletion of *CEP131* disrupted the centrosomal accumulation of CEP152 and CEP152 is required for the centrosomal localization of WDR62 and CEP63, we assessed whether the centrosomal localization of MCPH-associated proteins depends upon CEP131. The depletion of *CEP131* had no effect on the localization of CDK5RAP2, consistent with our finding that CEP152 is not required for CDK5RAP2 localization (*Figure 4G*). In addition to CEP152, depletion of *CEP131* abrogated the centrosomal localization, but not the stability, of the CEP152-dependent MCPH-associated proteins, WDR62 and CEP63 (*Figure 4G* and quantitated in *Figure 4—figure supplement 2A,B*). Together, these findings demonstrate that the centriolar satellite protein CEP131 is critical for bringing CEP152 to the centrosome, thereby allowing CEP152 to participate in centriole duplication (*Figure 4H*).

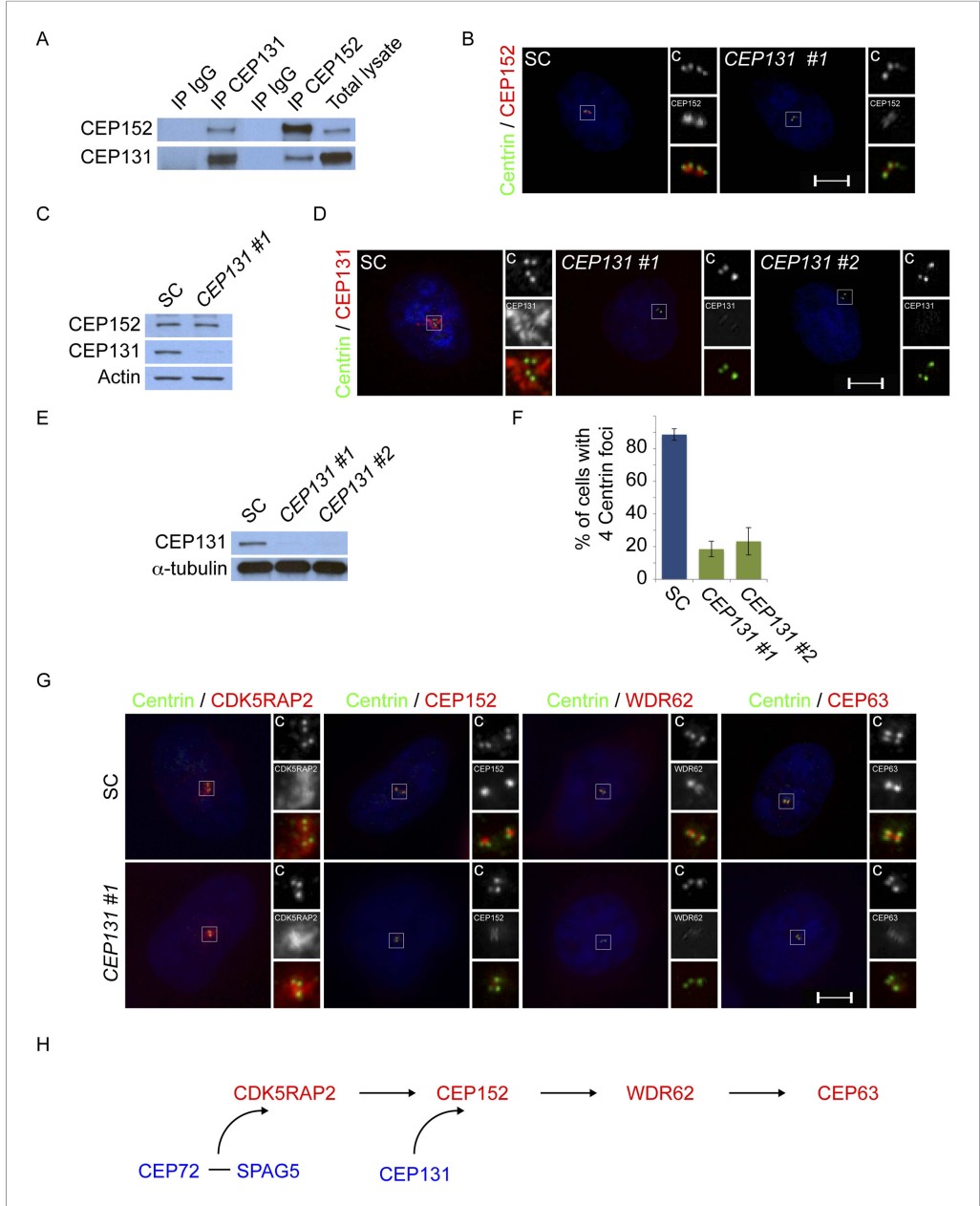

**Figure 4**. Centriolar satellite component CEP131 interacts with and localizes CEP152 to the centrosome to promote centriole duplication. (**A**) HeLa total cell lysates were subjected to immunoprecipitation of CEP131, CEP152 and a negative control. Precipitating proteins were subjected to immunoblotting for CEP131 and CEP152. (**B**) HeLa cells in S phase transfected with SC or *CEP131 #1* siRNA were co-stained for CEP152 (red) and Centrin ('c', green). (**C**) Total cell lysates of SC and *CEP131 #1* siRNA transfected HeLa cells were analyzed by immunoblotting with antibodies to CEP131 and CEP152. Actin served as a loading control. (**D**) HeLa cells in S phase transfected with SC, *CEP131 #1*, or *CEP131 #2* siRNA were co-stained for CEP131 (red) and Centrin ('c', green) to visualize centrioles. (**E**) Total cell lysates of SC, *CEP131 #1*, *CEP131 #2* siRNA transfected HeLa cells were analyzed by immunoblotting for Cep131. α-tubulin served as a loading control. 20 µg of protein lysate was loaded per lane. (**F**) Quantification of the mean percentage of SC, *CEP131 #1*, or *CEP131 #2* siRNA transfected cells in S phase with four centrioles. (**G**) SC and *CEP131*-depleted S phase cells were co-stained with Centrin ('c', green), CDK5RAP2 (red), CEP152 (red), WDR62 (red), and CEP63 (red). (**H**) Schematic indicating that CEP131 is required to localize CEP152, WDR62 and CEP63 to the centrosome. For all quantifications at least 100 cells were counted per experiment (n = 3), p < 0.005 (paired t-test). Scale bars indicate 5 µm for all images.

*Figure 4. continued on next page*

*Figure 4. Continued*

The following figure supplements are available for figure 4:

**Figure supplement 1**. *Cep131*^gt/gt^ MEFs exhibit centriole duplication and centrosome organizational defects.

**Figure supplement 2**. CEP131 is required for the centrosomal localization of CEP152, WDR62 and CEP63.

**Figure supplement 3**. CEP152 associates with CDK5RAP2 and WDR62 in a CEP131-dependent manner.

## Centriolar satellite protein MOONRAKER is required for MCPH-associated protein WDR62 to localize to the centrosome

Given the requirement of WDR62 for centriole duplication, we analyzed WDR62 co-precipitating proteins by mass spectrometry and identified MNR (also called KIAA0753), recently described as a centriolar satellite component involved in centriole duplication (*Jakobsen et al., 2011*; *Firat-Karalar et al., 2014*). We confirmed the interaction between MNR and WDR62 by reciprocal co-immunoprecipitation (*Figure 5A*). Immunoblotting and immunofluorescence revealed a loss of MNR in siRNA-treated lysates and cells (*Figure 5B,C*). We also confirmed that MNR is required to promote centriole duplication by depleting the protein and assessing centriole duplication and centrosome organization (*Firat-Karalar et al., 2014*). Depletion of *MNR* revealed that more than 73% of S phase cells had fewer than four centrioles (*Figure 5D*), a phenotype similar to that caused by the depletion of *WDR62* or *CEP63*.

Although overexpressed MNR interacts with CEP63 and MNR is essential for CEP63 localization to centrosomes (*Figure 5E* and *Figure 5—figure supplement 1A*, *Firat-Karalar et al., 2014*), we did not detect an interaction between endogenous MNR and CEP63 (data not included). As we had found that, instead, MNR interacts with WDR62 (*Figure 5A*), and WDR62 is required to localize CEP63 to the centrosome, we predicted that MNR would be required to localize WDR62 to centrosomes. Indeed, WDR62 was absent from centrosomes in *MNR*-depleted cells, indicating MNR is required for the centrosomal localization of WDR62 (*Figure 5F*).

As a centriolar satellite protein, CEP131, is required for CEP152 to localize to the centrosome and interact with other MCPH-associated proteins, we investigated whether, similarly, WDR62 required its cognate centriolar satellite protein to interact with the MCPH-associated proteins CEP152 and CEP63 at the centrosome. Therefore, we immunoprecipitated endogenous WDR62 from control and *MNR*-depleted cells, and found that, in the absence of MNR, WDR62 no longer co-precipitated CEP152 or CEP63 (*Figure 5—figure supplement 2A*), suggesting that assembling WDR62 together with other MCPH-associated proteins requires its centriolar satellite-mediated localization to the centrosome.

Similar to the relationship between CEP131 and CEP152, depletion of *MNR* did not destabilize its associated MCPH protein, WDR62 (*Figure 5G*). Consistent with a specific role in WDR62 localization, depletion of *MNR* did not affect CDK5RAP2 and CEP152 localization to the centrosome or its protein stability (*Figure 5H* and quantified in *Figure 5—figure supplement 1A,B*). Thus, loss of MNR phenocopies loss of WDR62, suggesting that this centriolar satellite component has a specific role in localizing WDR62 to the centrosome, which is subsequently required for the centrosomal localization of CEP63 and centriole duplication (*Figure 5I*).

## Centriolar satellite protein CEP90 is required for MCPH-associated protein CEP63 to localize to the centrosome

Cooperation between the MCPH-associated proteins culminates in CEP63 localization to centrosomes (*Figure 1*). As we had done with the other MCPH proteins, we subjected CEP63 and its co-immunoprecipitated proteins to analysis by mass spectrometry. Similar to other MCPH proteins, we identified a centriolar satellite component, CEP90, as a candidate interactor of CEP63 (*Supplementary file 1D*). Reciprocal immunprecipitation of endogenous proteins confirmed the interaction of CEP63 and CEP90 (*Figure 6A*).

We assessed whether CEP90 participated in the centrosomal localization of CEP63. Indeed, CEP63 was absent from the centrosome in *CEP90*-depleted cells (*Figure 6B* and quantified in

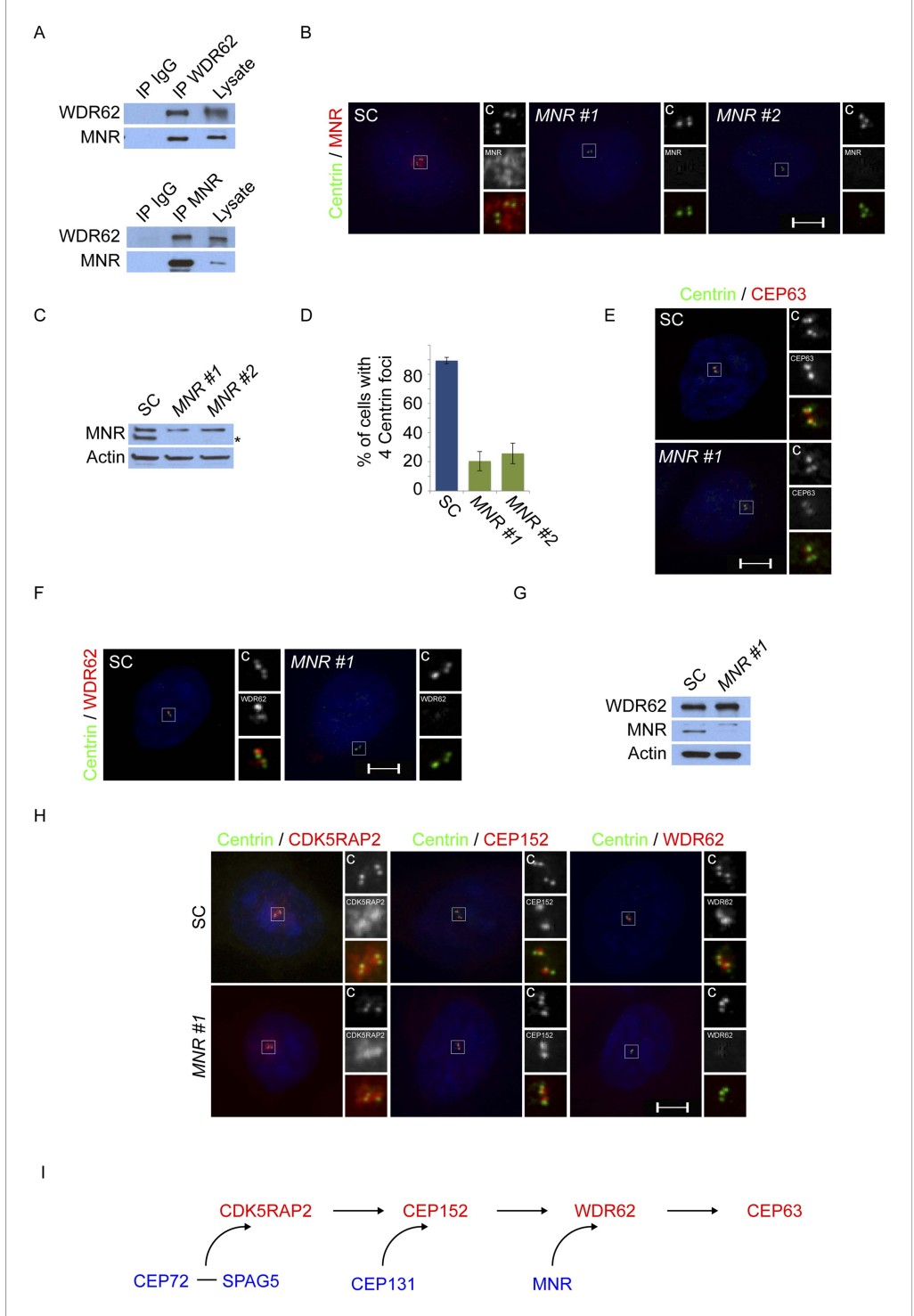

**Figure 5**. Satellite component MNR promotes centriole duplication by localizing WDR62 to the centrosome. (**A**) We immunoprecipitated endogenous WDR62 and MNR from HeLa total cell lysates. Co-precipitation was detected using antibodies specific to WDR62 and MNR. (**B**) SC, *MNR #1*, and *MNR #2* siRNA transfected S-phase HeLa cells were co-stained with MNR (red) and Centrin ('c', green). (**C**) Total cell lysate of SC, *MNR #1*, *MNR #2* siRNA transfected HeLa cells were analyzed by immunoblotting with antibodies to MNR. The asterisk marks the band specific to MNR, which sits below a non-specific band. Actin served as a loading control. (**D**) Percentage of S-phase SC, *MNR #1*, *MNR #2* siRNA treated HeLa cells with four Centrin foci. (**E**) S-phase SC and *MNR* siRNA-transfected HeLa cells were co-stained for Centrin ('c', green) and CEP63 (red). (**F**) SC and *MNR* siRNA treated S-phase HeLa

*Figure 5. continued on next page*

*Figure 5. Continued*

cells were co-stained for WDR62 (red) and Centrin ('c', green). (**G**) Total cell lysates of SC and *MNR*-depleted cells were analyzed by immunoblot with antibodies to WDR62 and MNR. Actin served as a loading control. 20 µg of protein lysate was loaded per lane. (**H**) SC and *MNR #1*-depleted S-phase cells were co-stained with Centrin ('c', green), CDK5RAP2 (red), CEP152 (red), and WDR62 (red). (**I**) Our findings indicate that MNR localizes WDR62 to the centrosome, which in turn recruits CEP63. For all quantifications at least 100 cells were counted per experiment (n = 3), p < 0.005 (paired t-test). Scale bars indicate 5 µm for all images.

The following figure supplements are available for figure 5:

**Figure supplement 1**. MNR is required to localize WDR62 and CEP63 to the centrosome.

**Figure supplement 2**. WDR62 interacts with CEP152 and CEP63 in a MNR-dependent manner.

---

*Figure 6—figure supplement 1A*), suggesting that CEP63 localizes to centrioles in a CEP90-dependent manner. The levels of CEP63 were unchanged upon depletion of *CEP90*, indicating that CEP90 is essential for CEP63 centrosomal localization, but not its stability (*Figure 6C*). Thus, each of the interacting MCPH-associated proteins has an associated centriolar satellite component that it requires for localization to centrosomes.

Given the requirement for other centriolar satellite proteins in assembling their cognate MCPH-associated proteins into a larger complex, we examined whether CEP90 is essential for CEP63 to associate with an interacting MCPH-associated protein, WDR62. We immunoprecipitated endogenous CEP63 from control and *CEP90*-depleted cells, and discovered that CEP90 is essential for CEP63 to co-precipitate WDR62 (*Figure 6—figure supplement 2A*). These results are consistent with the idea that centriolar satellite components bring select MCPH-associated proteins to the centrosome, and the MCPH-associated proteins form a complex specifically at the centrosome.

Previous studies have implicated CEP90 in ciliogenesis and mitotic spindle pole integrity (*Kim and Rhee, 2011*; *Kim et al., 2012*). Whether it has additional centrosomal functions has been unclear. We hypothesized that, like CEP63, CEP90 may also participate in centriole duplication. To investigate the role of CEP90 in centriole biogenesis, we depleted *CEP90* by siRNA and examined whether centriole duplication is abrogated. Immunofluorescence analysis demonstrated that greater than 78% of S phase *CEP90*-depleted cells had fewer than four centrioles (*Figure 6D–F*). These findings were similar to the centriole duplication defect we detected in *CEP63*-depleted cells, suggesting that CEP90 and CEP63 function together to promote centriole duplication.

As CEP63 is not required for the centrosomal localization of the other MCPH proteins CDK5RAP2, CEP152 and WDR62, we predicted that CEP90 would similarly be dispensable for their localization. Immunofluorescence and quantitative analysis of HeLa cells treated with *CEP90* siRNA revealed that the centrosomal localization and protein stability of CDK5RAP2, CEP152, and WDR62 were unaltered (*Figure 6G* and quantified in *Figure 6—figure supplement 1A,B*). Collectively, these results reveal that CEP63 localizes to the centrosome in a manner dependent on the centriolar satellite protein CEP90 and the MCPH proteins CDK5RAP2, CEP152 and WDR62 as well as the other centriolar satellite proteins CEP72, SPAG5, CEP131 and MNR (*Figure 6H*).

## A rare variant in CEP90 may be associated with microcephaly

Our finding that centriolar satellite proteins participate in the cellular functions of established MCPH-associated proteins raised the possibility that mutations in genes encoding centriolar satellite components might also cause MCPH. Examination of whole exome sequence from >300 families with MCPH identified a homozygous missense variant in *CEP90* in a consanguineous Pakistani family with two affected male siblings. The two affected boys (MC-701 and MC-702) were born to parents who were first cousins and who had a total of 12 pregnancies (*Figure 6I*). In addition to their two affected sons, they had four unaffected children, five miscarriages and one stillbirth at 6 months of gestation with multiple congenital anomalies. Both affected boys presented with a similar syndrome of microcephaly, intellectual disability, spasticity, and dysmorphisms. At birth, MC-701 measured 2.98 kg (18th percentile) for weight, 50 cm (48th percentile) for length and 33 cm (10th percentile) for head

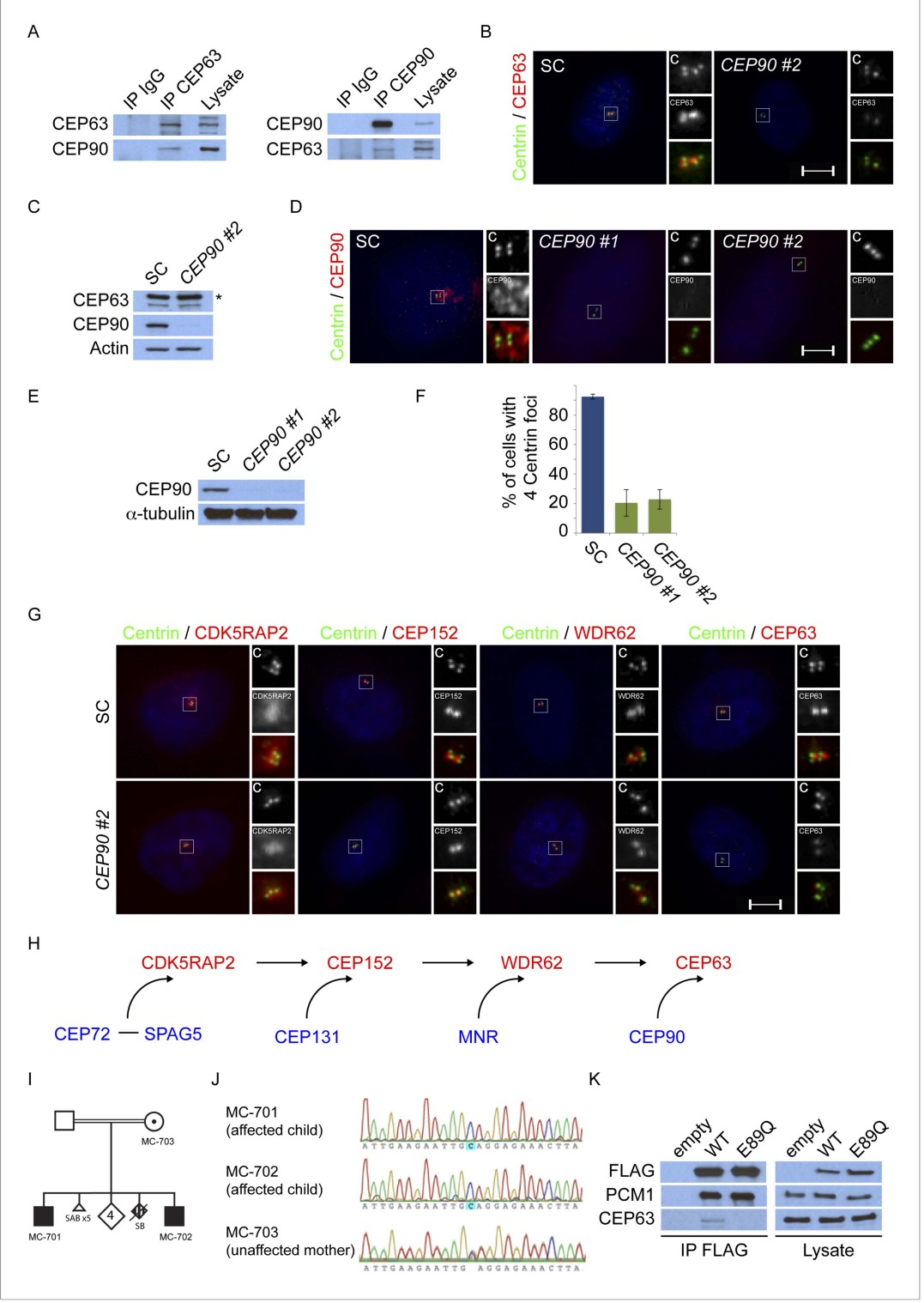

**Figure 6**. *CEP90* encodes a centriolar satellite and MCPH-associated protein required to localize CEP63 to the centrosome to promote centriole duplication. (**A**) HeLa total cell lysate was subjected to endogenous immunoprecipitation using antibodies to CEP63, CEP90 and a negative control. Precipitating proteins were analyzed by immunoblotting for CEP63 and CEP90. (**B**) Immunofluorescence images of SC and *CEP90 #2* siRNA-treated HeLa cells co-stained for CEP63 (red) and Centrin ('c', green). (**C**) Total cell lysates of SC or *CEP90 #2* siRNA transfected HeLa cells were analyzed by immunoblotting with antibodies to CEP63, CEP90 and Actin, which served as a loading control. Asterisk indicates the specific band. (**D**) S phase SC, *CEP90 #1*, and *CEP90 #2*

*Figure 6. Continued*

siRNA-treated cells were analyzed by immunofluorescence with antibodies to CEP90 (red) and Centrin ('c', green). (**E**) Total cell lysates of SC, *CEP90 #1*, and *CEP90 #2* siRNA transfected HeLa cells were analyzed by immunoblotting for CEP90. α-tubulin served as a loading control. (**F**) Mean percentage of S phase SC, *CEP90 #1*, and *CEP90 #2* siRNA-treated HeLa cells with four Centrin foci. At least 100 cells were counted per experiment (n = 3), p < 0.005 (paired t-test). (**G**) SC and *CEP90 #2*-depleted S phase cells were co-stained with Centrin ('c', green), CDK5RAP2 (red), CEP152 (red), WDR62 (red), and CEP63 (red). Scale bars indicate 5 μm for all images. (**H**) Schematic indicating that CEP90 is required for the centrosomal localization of CEP63. (**I**) Diagram of a simplified pedigree. Filled squares indicate individuals with MCPH. Spontaneous abortions (SAB), stillbirths (SB), and individuals of unknown gender (diamonds) are also indicated. (**J**) Sanger sequencing of the affected patients and their unaffected mother confirms the presence of a guanine to cytosine mutation leading to a charge reversing glutamic acid to glutamine substitution. This variant was identified by whole exome sequencing of the affected individuals and the unaffected mother. (**K**) We immunoprecipitated the FLAG tags of the wild-type and E89Q forms of CEP90 and blotted for endogenous CEP63 and FLAG.

The following figure supplements are available for figure 6:

**Figure supplement 1**. CEP90 is required to localize CEP63 to the centrosome.

**Figure supplement 2**. CEP63 interacts with WDR62 in a CEP90-dependent manner.

**Figure supplement 3**. *CEP90* human genetics.

---

circumference. Global developmental delay was noted at 5 months of age; he toe walked at 3 years, later developed tetraplegia, and is wheelchair-bound. He exhibits spasticity especially in the lower extremities with scissoring and Achilles contractures, hyperreflexia, ankle clonus and a positive Babinksi sign. He suffers from severe intellectual disability and does not speak. His dysmorphic features include hypertelorism, large ears, broad nose, anteverted nares, thick lips and a large mouth, as well as pectus carinatum and kyphoscoliosis. His brother, MC-702, has a similar phenotype with a head circumference of 48.5 cm (4 standard deviations below the mean) at 14 years of age. Brain magnetic resonance imaging noted normal gross morphology with atrophy.

Whole exome analysis suggested that *CEP90* is one of two strong candidates to contribute to the phenotype observed in this family. Both affected boys possess a homozygous missense alteration in the second coding exon of the *CEP90* gene (chr13:73366597G>C [hg19]), resulting in p.E89Q (*Figure 6J*). The observed variant lies in the midst of a 57 Mb stretch of homozygosity on chr13 implicated by linkage and homozygosity mapping to likely harbor the disease-causing allele (*Figure 6—figure supplement 3A–C*). *CEP90* p.E89Q has not been detected in dbSNP138, the 1000 Genomes Project, the ESP6500 databases, or >1000 exomes from 467 Middle Eastern families. Moreover, no other polymorphisms affecting E89 have been reported in these databases. E89 is highly conserved, including 9/9 vertebrate species including *Xenopus* and zebrafish as well as much more distantly related species (*Figure 6—figure supplement 3D*). The nonconservative p.E89Q alteration is strongly predicted by SIFT and Polyphen2 to be pathogenic (SIFT and Polyphen2 scores of 0 and 0.99, respectively). Both affected boys also have a homozygous intronic variant in *AP4S1* (rs185246578, NM_001254729:exon5:c.295-3C>A), a gene in which truncating mutations have been reported to cause a phenotype that overlaps that described in this family (*Hardies et al., 2015*). The variant detected in this family (rs185246578) is of uncertain significance, as it is an intronic SNP that was previously reported by the 1000 Genomes Project, is not strongly predicted to be pathogenic by splice prediction algorithms, and lies upstream of exon 5 of 6, which is skipped in one of the known *AP4S1* isoforms.

To test whether the observed *CEP90* mutation affects protein function, we engineered cDNAs corresponding to wild type and E89Q alleles and expressed the constructs in HeLa cells for 6 hr. We then assessed whether the mutant form of CEP90 was capable of interacting with its binding partners, PCM1 and CEP63 by co-immuoprecipitation. Interestingly, CEP90 E89Q interacted efficiently with PCM1, but did not bind CEP63 (*Figure 6K*). Thus, consistent with a role for CEP90 in the localization of the MCPH protein CEP63, our data suggests that mutations in CEP90 itself may contribute to abnormal human brain development.

## Centriolar satellite component CCDC14 restrains the centrosomal accumulation of CEP63 and CDK2

In addition to CEP90, our mass spectrometric analysis of CEP63 co-immunoprecipitating proteins identified CCDC14, another centriolar satellite component. It has previously been demonstrated that tagged versions of CEP63 bind CCDC14 (*Camargo et al., 2007*; *Firat-Karalar et al., 2014*). We confirmed that endogenous CEP63 and CCDC14 interact using reciprocal coimmunoprecipitions (*Figure 7A*). Immunoblot analysis demonstrated that *CCDC14* siRNAs efficiently depleted its target protein (*Figure 7B*). Unlike depleting *CEP90*, depleting *CCDC14* produced supernumerary Centrin foci in S phase U2OS or HeLa cells (*Figure 7C* and data not shown).

We assessed the composition of these Centrin foci in *CCDC14*-depleted HeLa cells by costaining with antibodies to γ-tubulin and CP110 to identify the pericentrosomal matrix and centriolar distal ends respectively. Interestingly, in *CCDC14*-depleted cells, Centrin foci colocalized with CP110 but only partially with γ-tubulin, suggesting that inhibiting CCDC14 cause the formation of supernumary Centrin foci not necessarily associated with pericentriolar matrices (*Figure 7—figure supplement 1A*).

A recent study demonstrated that CCDC14 restricts the centrosomal localization of CEP63 (*Firat-Karalar et al., 2014*). Consistent with these results, CEP63 localized to most of the supernumerary Centrin foci in *CCDC14*-depleted cells (*Figure 7D*). We confirmed by immunoblot that protein levels of CEP63 were unchanged in *CCDC14*-depleted cells (*Figure 7E*). In contrast to CEP63, CDK5RAP2, CEP152 and WDR62 were not recruited to these Centrin-positive foci, suggesting that the role for CCDC14 in restricting the localization of CEP63 to centrioles and centriole-like structures did not extend to other MCPH proteins (*Figure 7—figure supplement 2A* and quantified in *Figure 7—figure supplement 2B*). The stability of CDK5RAP2, CEP152, WDR62 and CEP63 was unaltered in *CCDC14*-depleted cells, indicating that the accumulation of CEP63 in *CCDC14*-depleted cells was not due to over-stabilization (*Figure 7—figure supplement 2C*).

As CEP63 depends on CDK5RAP2, CEP152 and WDR62 to localize to centrosomes, we assessed whether these other MCPH proteins were required for the formation of supernumerary Centrin foci in *CCDC14*-depleted cells. Using scrambled control or *CCDC14* siRNA together with siRNAs targeting *CDK5RAP2*, *CEP152*, *WDR62*, or *CEP63*, we depleted U2OS cells of CCDC14 in combination with MCPH-associated proteins (*Figure 7F*). Co-depletion of CCDC14 and MCPH proteins was confirmed by immunoblot analysis (*Figure 7—figure supplement 3*). Quantitation of Centrin foci revealed that, whereas depleting CCDC14 alone caused the formation of supernumary Centrin foci, co-depleting *CDK5RAP2*, *CEP152*, *WDR62* or *CEP63* blocked the formation of supernumary Centrin foci (*Figure 7G*). In contrast, co-depleting the MCPH-associated proteins produced a phenotype similar to depleting the MCPH-associated protein alone. These findings demonstrate that CCDC14 suppresses the formation of supernumerary Centrin foci in a manner that depends upon the MCPH-associated proteins CDK5RAP2, CEP152, WDR62 and CEP63.

CEP63 interacts with Cyclin dependent kinase 2 (CDK2), a cell cycle-dependent kinase that is activated at the onset of S phase and participates in centriole duplication (*Dulic et al., 1992*; *Hinchcliffe et al., 1999*; *Lacey et al., 1999*; *Matsumoto et al., 1999*; *Meraldi et al., 1999*; *De Boer et al., 2008*; *Loffler et al., 2011*). Reciprocal immunoprecipitations of CEP63 and CDK2 confirmed the specificity of the interaction (*Figure 7—figure supplement 4A*). To assess whether the localization of CDK2 was altered in the absence of the CEP63-interacting protein CCDC14, we assessed the localization of CDK2 in *CCDC14*-depleted cells. CDK2, which localizes in the vicinity of the centrosome, overaccumulated at the Centrin foci in *CCDC14*-depleted cells (*Figure 7H*).

As CDK2 activity is required for centriole duplication and its inhibition can suppress supernumerary Centrin foci induced by CEP63 overexpression (*Loffler et al., 2011*), we assessed whether blocking CDK2 activity could similarly suppress the formation of supernumerary Centrin foci in *CCDC14*-depleted cells. We examined U2OS cells transfected with scrambled control (SC) or *CCDC14* siRNA and treated with DMSO or the CDK2 inhibitor roscovitine. Following DMSO treatment, control cells in S phase had two pairs of centrioles while *CCDC14*-depleted cells had multiple Centrin foci (*Figure 7I*, left, and *Figure 7J*). In contrast, *CCDC14*-depleted cells in S phase treated with roscovitine only contained one pair of centrioles (*Figure 7I*, right, and *Figure 7J*). Together, these findings demonstrate that CCDC14 suppresses the formation of supernumerary Centrin foci by limiting the centrosomal localization of CEP63 and the activity of its binding partner CDK2.

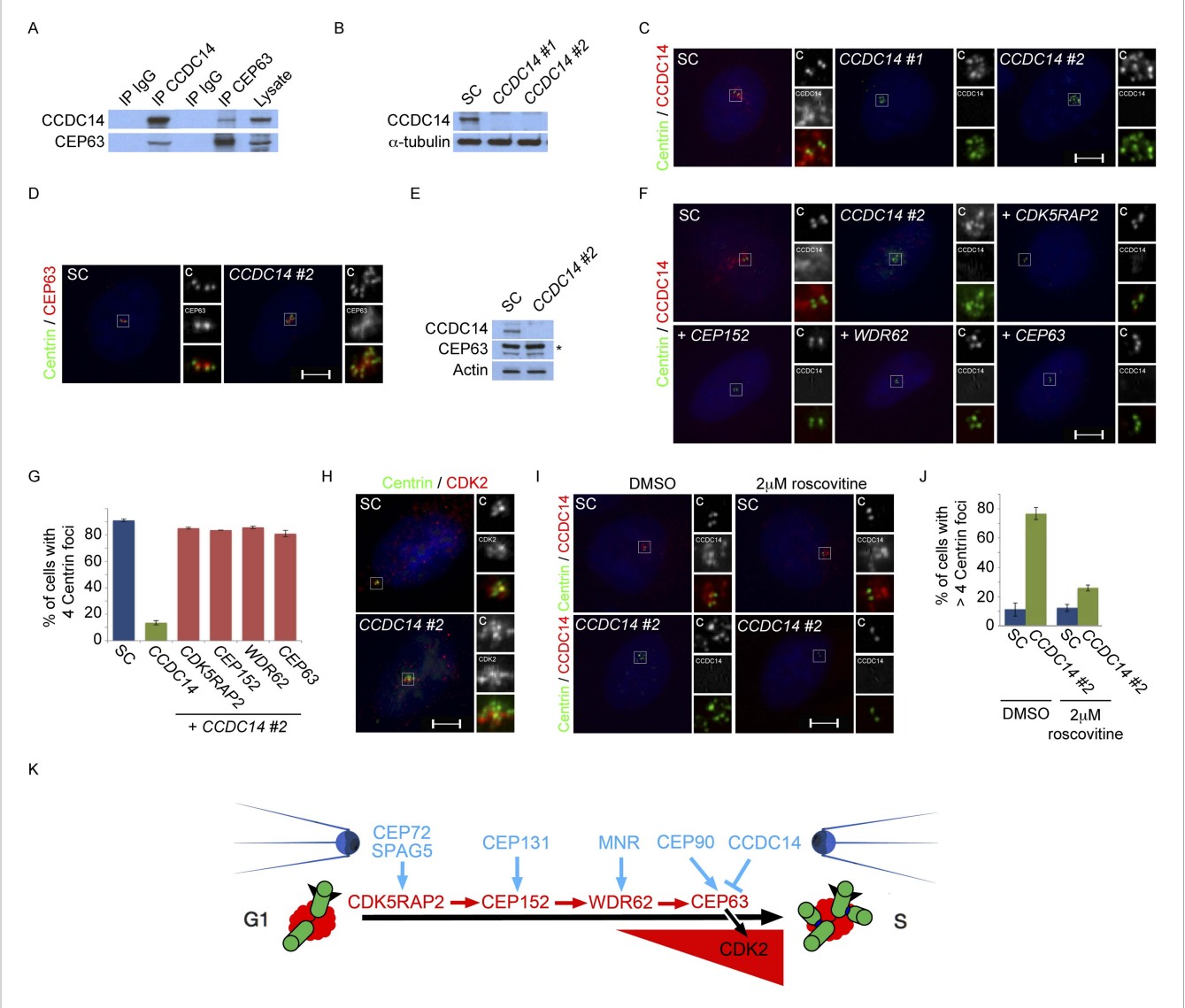

**Figure 7**. Satellite component CCDC14 suppresses the formation of supernumerary Centrin foci by limiting the centrosomal accumulation of CEP63 and activity of CDK2. (**A**) We immunoprecipitated endogenous CCDC14 and CEP63 from HeLa total cell lysates. Efficient precipitation and co-precipitation were detected using antibodies specific to CCDC14 and CEP63. (**B**) S phase SC, *CCDC14 #1*, and *CCDC14 #2* siRNA transfected U2OS cells were analyzed by immunofluorescence with antibodies to CCDC14 (red) and Centrin ('c', green). (**C**) Total cell lysates of SC, *CCDC14 #1*, and *CCDC14 #2* siRNA-treated U2OS cells were analyzed by immunoblotting for CCDC14. α-tubulin served as a loading control. (**D**) S phase SC and *CCDC14 #2* siRNA-treated U2OS cells were co-stained for Centrin ('c', green) and CEP63 (red). (**E**) Total cell lysate of SC, *CCDC14 #2* siRNA transfected U2OS cells were analyzed by immunoblotting with antibodies to CCDC14 and CEP63. Actin served as a loading control. (**F**) U2OS cells transfected with SC siRNA, or siRNA targeting *CCDC14* alone, or *CCDC14* siRNA in combination with siRNAs targeting *CDK5RAP2, CEP152, WDR62, or CEP63* in S phase were co-stained with Centrin ('c', green) and CCDC14 (red). (**G**) Quantification of S phase SC, *CCDC14*, or *CCDC14* and *CDK5RAP2, CEP152, WDR62, or CEP63* siRNA-treated U2OS cells with fewer than four centrioles. S phase cells were identified by Cyclin A immunostaining. For all quantifications at least 100 cells were counted per experiment (n = 3), p < 0.005 (paired t-test). (**H**) S phase SC and *CCDC14 #2* siRNA transfected U2OS cells were co-stained with Centrin ('c', green) and CDK2 (red). (**I**) S phase SC and *CCDC14 #2* siRNA transfected U2OS cells treated with DMSO or roscovitine were analyzed by immofluorescence with antibodies to CCDC14 (red) and Centrin ('c', green). Scale bars indicate 5 μm for all images. (**J**) Percentage of S phase SC, *CCDC14 #2* siRNA-transfected U2OS cells treated with DMSO or roscovitine with greater than four Centrin foci. (**K**) CDK5RAP2 delivery to the centrosome requires the centriolar satellite proteins CEP72 and SPAG5. In a manner dependent on CDK5RAP2, CEP131 localizes CEP152 to the centrosome, MNR localizes WDR62, and CEP90 localizes CEP63. The centriolar satellite, CCDC14 binds CEP63 and removes it from the centrosome to limit centriole duplication by limiting the localization and activity of CDK2.

*Figure 7. continued on next page*

*Figure 7. Continued*

The following figure supplements are available for figure 7:

**Figure supplement 1**. CCDC14 limits the formation of Centrin positive foci that do not recruit the PCM component γ-tubulin.

**Figure supplement 2**. CCDC14 limits the centrosomal accumulation of CEP63 but does not alter CDK5RAP2, CEP152, or WDR62 localization.

**Figure supplement 3**. Co-depletion of CCDC14 and MCPH-associated proteins.

**Figure supplement 4**. CEP63 and CDK2 interact.

**Figure supplement 5**. CDK2 localizes to the centrosome in a CDK5RAP2, CEP152, WDR62, CEP63, SPAG5, CEP72, CEP131, MNR, and CEP90-dependent manner.

**Figure supplement 6**. CDK5RAP2, CEP152, WDR62, CEP63, SPAG5, CEP72, CEP131, MNR, CEP90 and CCDC14 do not control the stability of CDK2.

Given the involvement of CDK2 in promoting centriole duplication, we examined whether the centrosomal localization of CDK2 was dependent on MCPH proteins and their cognate centriolar satellite interactors that promote centriole duplication. Immunostaining U2OS cells treated with siRNA to *CDK5RAP2, CEP152, WDR62, CEP63, SPAG5, CEP72, CEP131, MNR,* or *CEP90* revealed that all of these MCPH-associated genes and centriolar satellite genes promote the centrosomal localization of CDK2 (*Figure 7—figure supplement 5A* and quantified in *Figure 7—figure supplement 5B*). The protein level of CDK2 was unaltered in these MCPH and centriolar satellite protein knockdown cells (*Figure 7—figure supplement 6A*). Collectively, these results reveal that CDK2 localizes to the centrosome in a manner dependent on MCPH-associated proteins and select centriolar satellite proteins to promote centriole duplication (*Figure 7K*).

## Discussion

We found that the MCPH-associated proteins CDK5RAP2, CEP152, WDR62 and CEP63 interact with each other and function together to promote centriole duplication. These MCPH-associated proteins assemble in a step-wise hierarchical manner to form a toroidal ring at the proximal centriole. We also found that each of these MCPH-associated proteins interacts with distinct centriolar satellite components that are required for the ordered recruitment of its cognate MCPH protein to the centrosome. Therefore, the centriolar satellite proteins, like the MCPH proteins, have critical roles in promoting centriole duplication. Consistent with a role of centriolar satellites in the function of proteins previously associated with MCPH, we have found evidence that the centriolar satellite gene, *CEP90*, may be mutated in rare instances of MCPH.

The assembly of MCPH proteins culminates in CDK2 localization to centrosomes, a previously recognized regulator of centriole duplication (*Hinchcliffe et al., 1999*; *Matsumoto et al., 1999*). In contrast to the other centriolar satellites components that promote MCPH-associated protein localization to the centrosomes and centriole duplication, CCDC14 opposes this pathway by limiting the centrosomal localization of CEP63 and its interactor CDK2. Thus, centriolar satellite proteins can have either positive or negative roles in controlling the centrosomal localization of MCPH-associated proteins and CDK2, and are thus critical for centriole duplication (*Figure 7K*).

Several previous studies in various organisms have demonstrated how proteins can recruit each other to the centrosome to promote centriole duplication. For example, the *Drosophila melanogaster* homologs of CDK5RAP2 and CEP152, Centrosomin and Asterless, interact with each other and Asterless is involved in the centrosomal localization of Centrosomin (*Conduit et al., 2010*). In *C. elegans,* centriole duplication is coordinated by SPD-2 (CEP192) recruitment of ZYG-1 (PLK4) which, in turn, localizes SAS-5 (STIL) and SAS-6 to the centrosome, culminating in the recruitment of SAS-4 (*Delattre et al., 2006*). Our results indicate that ordered recruitment of MCPH-associated proteins to the centrosome extends beyond CDK5RAP2 and CEP152 to include WDR62 and CEP63. Moreover, the recruitment of centriole duplication proteins to the centrosome occurs in a step-wise fashion, with CDK5RAP2 being required for the recruitment of CEP152, which recruits WDR62, which in turn recruits CEP63 (*Figure 7J*).

While our mass spectrometry analysis of CEP152 detected interactions with CDK5RAP2, WDR62 and CEP131, we did not detect some previously reported interactors of CEP152, including CEP192,

SAS4, CENTROBIN and PLK4 (*Cizmecioglu et al., 2010*; *Hatch et al., 2010*; *Kim et al., 2013*; *Sonnen et al., 2013*; *Firat-Karalar et al., 2014*; *Gudi et al., 2014*). Immunoprecipitation of endogenous CEP152 may be less efficient than overexpressed CEP152, or the use of different cell lines or lysis conditions may account for the observed differences. Conversely, our in solution mass spectrometry preparations allowed us to detect an interaction between CEP152 and WDR62. These interaction data helped reveal that WDR62 localizes to the centrosome in a CEP152-dependent manner.

Interestingly, although CEP63 has been reported to be required for the localization of CEP152 to the centrosome, we detected only a minor decrease in the centrosomal accumulation of CEP152 in *CEP63*-depleted cells 48 hr post-transfection (*Sir et al., 2011*; *Brown et al., 2013*; *Lukinavicius et al., 2013*). The observed difference in dependence of localization is attributable to the loss of CEP152 protein stability upon knockdown of *CEP63* using the siRNA sequence used by Brown et al. The additional *CEP63* siRNAs used in this study efficiently deplete CEP63 but do not affect CEP152 protein levels (*Figure 1—figure supplement 5*), suggesting that CEP152 centrosomal localization does not critically depend on CEP63.

The MCPH-associated protein CEP63 interacts with CDK2, a kinase which in addition to its well known roles in cell cycle progression, is also involved in centriole duplication (*Hinchcliffe et al., 1999*; *Matsumoto et al., 1999*; *Loffler et al., 2011*). Consistent with a key role for CEP63 in recruiting CDK2, the other MCPH-associated proteins CDK5RAP2, CEP152 and WDR62 required for CEP63 localization to the centrosome are also required for CDK2 localization to the centrosome and centriole duplication. While the recruitment of CDK2 to the centrosome by MCPH-associated proteins can account for how these proteins participate in centriole duplication, our results do not exclude roles for the MCPH proteins in recruiting other effectors of centriole duplication to the centrosome.

We also found that the centriolar satellite components SPAG5, CEP72, CEP131, MNR, and CEP90 participate in centriole duplication by interacting with and localizing MCPH-associated proteins to the centrosome. Different satellite proteins interact with different MCPH proteins: CEP72 and SPAG5 interact with CDK5RAP2, CEP131 with CEP152, MNR with WDR62, and CEP90 with CEP63. By bringing MCPH-associated proteins to the centrosome, the centriolar satellite proteins are required to localize proteins to the centrosome even when no biochemical interaction was detected. For example, CEP72 and SPAG5 are required for all four MCPH-assocated proteins to localize to the centrosome, but they interact specifically with CDK5RAP2. While it is possible that the satellite proteins assemble the MCPH protein complex in the cytoplasm prior to its delivery to the centrosome, the most parsimonious model to explain these results is that centriolar satellite proteins bring individual interacting MCPH-associated proteins to the centrosome where the MCPH complex is assembled, and that this recruitment and assembly is required for the subsequent localization of other MCPH-associated proteins farther down the hierarchy. To extend our example, by bringing CDK5RAP2 to the centrosome, the centriolar satellite proteins CEP72 and SPAG5 are required for the centrosomal localization of the other three MCPH proteins despite not interacting with them biochemically. Thus, the centriolar satellite proteins form a hierarchy that parallels the MCPH protein hierarchy, with distinct satellite components interacting with and being required for the centrosomal localization of different MCPH-associated proteins.

Inhibiting centriolar satellite protein function not only prevents the interacting MCPH-associated protein from localizing to the centrosome, but it also prevents that MCPH protein from interacting with the other MCPH-associated proteins, indicating that it is critical for MCPH complex formation. These findings suggest that centriolar satellite proteins may bind their cognate MCPH-associated protein outside of the centrosome and deliver it to the centrosome where it can complex with other MCPH-associated proteins. Previous studies revealed that centriolar satellites transport cargo along microtubules in a dynein-dynactin dependent manner to deliver proteins to the centrosome (*Kubo et al., 1999*; *Kodani et al., 2010*). Our finding that MCPH-associated proteins colocalize with the centriolar satellite protein PCM1 upon acute depolymerization of microtubules provides further evidence that the function of centriolar satellites is to transport proteins to the centrosome in a microtubule-dependent manner (*Figure 2—figure supplement 1*, *Dammermann and Merdes, 2002*; *Kodani et al., 2010*). Thus, a critical function of centriolar satellite proteins may be to bring MCPH-associated proteins to the centrosome, thereby accumulating CDK2 and promoting centriole duplication. Recent evidence implicating centriolar satellites in various cellular processes such as cellular stress response, ciliogenesis and protein degradation, reveals that centriolar satellite function extends beyond promoting centriole duplication (*Kim et al., 2012*; *Stowe et al., 2012*; *Hall et al., 2013*; *Villumsen et al., 2013*).

We identified a homozygous missense variant in the centriolar satellite gene, *CEP90,* in two boys with microcephaly and intellectual disability. The charge-reversing mutation affects a glutamate residue conserved back to protists and eliminates the interaction of CEP90 with the MCPH protein CEP63, but not its interaction with PCM1. Therefore, we propose that hypomorphic mutations in *CEP90* are a rare cause of human microcephaly, and that defects in the function of centriolar satellites can cause the human disease microcephaly. It will be interesting to determine whether mutations in genes encoding other centriolar satellite components are also causes of microcephaly.

Depletion of *CEP90* reduces recruitment of CEP63 and CDK2 to the centrosome and abrogates centriole duplication. We hypothesize that microcephaly-associated mutation of *CEP90* causes defective centriole duplication, similar to depletion of CEP90 or of MCPH proteins such as CDK5RAP2, CEP152, WDR62 or CEP63. Loss of function mutations in the mouse orthologs of the MCPH genes *MCPH1, STIL, CDK5RAP2,* and *ASPM* alter spindle orientation and promote the loss of neural progenitors (*Izraeli et al., 1999*; *Barrera et al., 2010*; *Lizarraga et al., 2010*; *Pulvers et al., 2010*; *Gruber et al., 2011*). It seems likely that centriole duplication is critical for spindle orientation and the regulation of neuronal progenitor number, but why centriole duplication is preferentially required for brain development remains unclear.

In contrast to the other MCPH protein-interacting satellite components, CCDC14 restrains supernumerary Centrin foci formation by limiting the centrosomal accumulation of its interacting protein CEP63 (*Firat-Karalar et al., 2014*). We found that CCDC14 is also essential for limiting the centrosomal localization and activity of CDK2. As several satellite proteins are required for MCPH protein localization to the centrosome, but CCDC14 limits the centrosomal localization of CEP63 and CDK2, centriolar satellites may control the trafficking of proteins both to and away from the centrosome.

In summary, we have found that diverse centriolar satellite proteins control the localization of cognate MCPH-associated proteins to the centriole, where they assemble in a hierarchical step-wise manner, culminating in the recruitment CDK2 to the centrosome and centriole duplication. These findings help elucidate how satellites and ordered assembly coordinately build the complex centrosomal structure and how MCPH-associated proteins participate in the fundamental cell biological process of centriole duplication.

## Materials and methods

### Antibodies

Polyclonal guinea pig antibodies to human CEP131 (aa. 1053–1083) and CCDC14 (aa. 194–211) were developed by Pierce Technologies (Thermo Fisher Scientific Inc., Rockford, IL). Antibodies obtained for this study are the following: anti-Centrin1/2 20H5 (Millipore, Billerica, MA), anti-CDK5RAP2 (ABE236 and 06–0398, Millipore, Billerica, MA), anti-CEP63 (Millipore, Billerica, MA, Proteintech, Chicago, IL and Thermo Fisher Scientific, Waltham, MA), anti-MNR (Novus Biologicals, Littleton, CO and Sigma, St. Louis, MO), anti-CCDC14 (GeneTex Inc., Irvine, CA and Abcam, Cambridge, MA), anti-CEP131 (Sigma Prestige, St. Louis, MO and Abcam, Cambridge, MA), anti-CP110, anti-CEP72, anti-SPAG5, anti-CEP90, anti-Centrin1 (Proteintech Group, Chicago, IL), anti-PCM1 (H-262 and D-19, Santa Cruz Biotechnologies, Dallas, TX), anti-CDK2 (M2, Santa Cruz Biotechnology, Dallas, TX; 22060-1-AP, Proteintech Group, Chicago, IL), anti-WDR62 and anti-CEP152 (Bethyl Labs, Rockford, IL), anti-Actin, anti-α-tubulin and anti-γ-tubulin (Sigma, St. Louis, IL), anti-SAS6 and anti-c-Myc (Santa Cruz Biotechnology, Dallas, TX). Alexa-conjugated secondary antibodies and Hoechst 33342 were obtained from Molecular Probes (Life Technologies, Grand Island, NY). DyLight and HRP-conjugated secondary antibodies were obtained from Jackson ImmunoResearch Laboratories (West Grove, PA) and Cell Signaling (Grand Island, NY).

### Cell culture and siRNA transfections

HeLa and U2OS cells (UCSF tissue culture facility) were cultured in Advanced DMEM (Invitrogen, Grand Island, NY) supplemented with 2% FBS (Invitrogen, Grand Island, NY) and Glutamax-I (Invitrogen, Grand Island, NY). Lyophilized roscovitine (Millipore, Billerica, MA) was resuspended in DMSO. HeLa cells were treated with 17 μM of nocodazole for 45 min to depolymerize microtubules. To inhibit CDK2 function, U2OS cells were incubated with 2 μM of roscovitine for 24 hr. Cells were transfected with siRNA using Oligofectamine (Invitrogen, Grand Island, NY) according to the manufacturer's instructions and analyzed 48 hr later. Sequences for the siRNA oligonucleotides used in this study are described in *Supplementary file 2*. To deplete *CEP215* and *CEP63* using previously

published sequences, HeLa cells were grown in DMEM (Invitrogen, Grand Island, NY) supplemented with 10% FBS, and Glutamax-I.

## Immunoprecipitation and immunoblotting

Asynchronous HeLa cells were incubated on ice for 5 min in chilled $Ca^{++}$ and $Mg^{++}$ free Dulbecco's PBS (DPBS, Invitrogen, Grand Island, NY), harvested using a cell scraper and lysed on ice for 10 min in lysis buffer (50 mM Tris-HCL pH7.4, 266 mM NaCl, 2.27 mM KCl, 1.25 mM $KH_2PO_4$, 6.8 mM $Na_2HPO_4$-$7H_2O$ and 1% NP-40) supplemented with protease and phosphatase inhibitors (Calbiochem, Billerica, MA and Thermo Fisher Scientific Inc., Waltham, MA). Lysates were clarified (13,000 r.p.m., 4°C, 10 min) and the protein concentrations were determined using the Bradford assay (Bio-Rad, Hercules, CA). For each immunoprecipitation reaction analyzed by mass spectrometry, 5 mg of total lysate was incubated with 10 µg of antibody for 2 hr and then incubated with protein G-sepharose (GE Healthcare Life Sciences, Pittsburgh, PA) for an additional 1.5 hr at 4°C. Immunocomplexes were washed three times in lysis buffer, once in detergent free lysis buffer and subsequently incubated with 0.2 M Glycine pH2.5 on ice for 10 min and quenched with 1 M Tris pH9. For smaller scaled immunoprecipitation reactions, 500 µg of total lysate was incubated with 2 µg of antibody for 2 hr and then incubated with protein G-Sepharose for an additional hour at 4°C. Complexes were washed three times in lysis buffer and subsequently boiled in 2× Laemmli reducing buffer. To detect WDR62, samples were incubated in 2× Laemmli reducing buffer, but were not boiled. Samples were separated on 4–15% gradient TGX precast gels (Bio-Rad, Hercules, CA), transferred onto nitrocellulose (Whattman, Pittsburgh, PA) and then subjected to immunoblot analysis using ECL Lightening Plus (Perkin–Elmer, Waltham, MA). For quantifications, samples were blotted with fluorescently labeled secondary antibodies and analyzed on a LiCOR Odyssey scanner. Specific protein signals were normalized to Actin.

## Centrosome isolation

Centrosomes from asynchronously growing HeLa cells were isolated as previously described (*Bobinnec et al., 1998*). HeLa cells were treated with 2 µM nocodazole and 1 µg/µl of cytochalasin D for 1 hr to disrupt the microtubule and actin cytoskeleton respectively. Cells were collected and lysed in lysis buffer (1 mM HEPES, pH 7.2, 0.5% NP-40, 0.5 mM $MgCl_2$, and 0.1% β-mercaptoethanol) and clarified at 2500×g for 10 min. The resulting supernatant was layered atop a 60% sucrose cushion above a discontinuous sucrose gradient (70, 50 and 40% sucrose) and centrifuged at 40,000×g for 1 hr. Fractions were collected from the bottom and analyzed by Western blot.

## Immunofluorescence

To visualize centrosome and centriolar satellite proteins, cells were fixed in chilled methanol for 3 min. Following fixation; cells were incubated in blocking buffer (2.5% BSA, 0.1% Triton-X100, 0.03% $NaN_3$ in DPBS) overnight at 4°C. Primary and secondary antibodies were diluted in blocking buffer and incubated with cells at room temperature for 1 hr. To detect CDK2 using anti-CDK2 (M2, Santa Cruz Biotechnology), cells were extracted with 0.1% Triton X-100 in a buffer consisting of 50 mM piperazine-1, 4-bis (2-ethanesulphonic acid) at pH 7.4, 5 mM $MgCl_2$, and 5 mM EDTA prior to fixation in chilled methanol for 3 min. Alternatively, cells were fixed in ice cold methanol:acetone (1:1) for 7 min to detect CDK2 using anti-CDK2 (22060-1-AP, Proteintech Group, Chicago, IL). Coverslips were mounted using Gelvatol mounting media and imaged with an inverted Axio Observer D1 (Zeiss, Thornwood, NY), image processing was completed with Adobe Photoshop. For fluorescence quantifications, specified sized regions of interest were selected and quantified using Fiji. The signal of the MCPH proteins were normalized to Centrin signal in HeLa or U2OS cells.

## Mass spectrometry analysis

To analyze proteins by LC-MS/MS, immunocomplexes were digested with trypsin (Promega, Madison, WI) overnight at 37°C, denatured in 2 M urea, 10 mM $NH_4HCO_3$, 2 mM DTT for 30 min at 60°C, and alkylated with 2 mM iodoacetamide for 45 min at room temperature. Samples were analyzed by LC-MS/MS with a Thermo Scientific Velos Pro ion trap mass spectrometry system equipped with a Proxeon Easy nLC 1000 ultra high pressure liquid chromatography and autosampler system. Peptide

data was matched to protein sequences by the Protein Prospector algorithm and searched against the SwissProt Human protein sequence database.

## Human subjects and genetics

Subjects were identified and evaluated in a clinical setting for medical history, cognitive impairment and physical abnormalities. Those with histories suggestive of hereditary disorders of brain development, epilepsy, and/or cognition were offered research enrollment. After obtaining written informed consent from participants or their legal guardian, phenotypic information and peripheral blood samples were collected for research purposes. The informed consent for this genetic study included permission for publication. Ethical review and approval was obtained from the Committee on Clinical Investigations at Beth Israel Deaconess Medical Center (current protocol 2001-P-000758). The study adheres to the state and federal regulations governing the conduct of human subject research (45 CFR Part 46 and 21 CFR Parts 50 and 56) and the ethical principles set forth in the Belmont Report.

Patient DNA samples were subject to genome-wide SNP genotyping using Illumina 660W-Quad BeadChip (2.6M markers). Linkage analyses were performed using Allegro, MERLIN and homozygosity analysis was performed using custom scripts as previously described (*Yu et al., 2010*). Whole exome sequencing libraries were generated using Agilent SureSelect capture kits, and sequenced on Illumina HiSeq machines to a mean read depth of 170×. Alignment, variant calling, and annotation were performed as previously described (*Yu et al., 2013*).

## Plasmids

3X-FLAG tagged human *CEP90* cDNA was obtained from Genecopoeia (EX-T9157-M12). The single–base pair mutation in CEP90 was introduced using the QuickChange site-directed mutagenesis kit (Agilent Technologies) with the primer pair: Forward: 5′-cttacaaagattgaagaattggacgagaaacttaatgatgcacttca-3′; Reverse: 5′-tgaagtgcatcattaagtttctcgtccaattcttcaatctttgtaag-3′.

## Acknowledgements

We thank Drs Tim Stearns, Laurence Pelletier, and Brian Dynlacht for helpful discussions. This work was funded by grants by the NIH (T32HL007731) and the Sandler Family Supporting Foundation (PBBR) to ATK, the NIH (AR054396, GM095941), the Burroughs Welcome Fund, the Packard Foundation, and the Sandler Family Supporting Foundation to JFR.

---

## Additional information

### Funding

| Funder | Grant reference | Author |
| --- | --- | --- |
| National Institute of Arthritis and Musculoskeletal and Skin Diseases (NIAMS) | AR054396 | Jeremy F Reiter |
| National Institutes of Health (NIH) | P50 GM081879 | Nevan J Krogan |
| Burroughs Wellcome Fund (BWF) | | Jeremy F Reiter |
| National Institute of General Medical Sciences (NIGMS) | GM095941 | Jeremy F Reiter |
| Sandler Foundation | Program for Breakthrough Biomedical Research (PBBR) | Andrew Kodani |
| National Heart, Lung, and Blood Institute (NHBLI) | T32HL007731 | Andrew Kodani |
| National Heart, Lung, and Blood Institute (NHBLI) | P01 HL089707 | Nevan J Krogan |

The funders had no role in study design, data collection and interpretation, or the decision to submit the work for publication.

## Author contributions

AK, Conception and design, Acquisition of data, Analysis and interpretation of data, Drafting or revising the article; TWY, Acquisition of data, Analysis and interpretation of data, Drafting or revising the article; JRJ, Acquisition of data, Analysis and interpretation of data; DJ, NJK, Acquisition of data, Contributed unpublished essential data or reagents; TLJ, JNP, HK, ALK, Contributed unpublished essential data or reagents; LA-G, LS, Provided genetic information for human subjects, Acquisition of data; AD, CAW, Drafting or revising the article, Contributed unpublished essential data or reagents; JFR, Analysis and interpretation of data, Drafting or revising the article

## Ethics

Human subjects: Subjects were identified and evaluated in a clinical setting for medical history, cognitive impairment and physical abnormalities. Peripheral blood samples were collected from the affected individuals and family members after obtaining written informed consent according to the protocols approved by the participating institutions and the ethical standards of the responsible national and institutional committees on human subject research.

# Additional files

### Supplementary files

• Supplementary file 1. CDK5RAP2, CEP152, WDR62 and CEP63 mass spectrometry analysis. (**A–D**) Selected list of CDK5RAP2, CEP152, WDR62, and CEP63 interacting centrosomal proteins, including peptide counts and percent coverage. Coprecipitating proteins were counterscreened against c-Myc and FLAG interacting proteins, which served as negative controls. We used the proteomic analysis by *Andersen et al. (2003)* and *Jakobsen et al. (2011)* to suggest the identity of centrosomal proteins for additional analysis. Precipitated proteins are highlighted in blue, MCPH-associated proteins are in green, and centriolar satellite proteins are in orange.

• Supplementary file 2. siRNA sequences. Human Stealth siRNAs to *CDK5RAP2, WDR62, CEP63, SPAG5, CEP72, CEP131, MNR, CEP90* and *CCDC14* were obtained from Life Technologies. SC, *CEP152, CEP63 NB* and *CEP215* siRNAs were synthesized by Life Technologies.

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
