## [Decision Letter]

Thank you for sending your work entitled “Centriolar satellites bring microcephaly-associated proteins to centrosomes to promote centriole duplication” for consideration at *eLife*. Your article has been favorably evaluated by Vivek Malhotra (Senior editor), a Reviewing editor, and three reviewers.

The Reviewing editor and the other reviewers discussed their comments before we reached this decision, and the Reviewing editor has assembled the following comments to help you prepare a revised submission.

This work by Kodani and colleagues from the Reiter lab explored interactions between different microcephaly-associated proteins and between microcephaly-associated proteins and centriolar satellite proteins. The authors seek to make the point that centriolar satellites are crucial for centrosome biogenesis and functionally link satellite proteins to MCPH. The paper is descriptive and functional studies are at the level of dependence of one protein for the localization of another. In spite of the title there is no data showing that satellites bring microcephaly proteins to centrosomes. The paper is difficult to read and will be a challenge for anyone not directly involved in this field. Perhaps a model or diagram would help to graphically illustrate the data.

Major points:

1) The hierarchical centrosome recruitment model has not been established thoroughly:

A) The authors showed that depletion of centriolar satellite proteins SPAG5, CEP72, CEP131, MNR, and CEP90 disrupted centriole duplication. They claimed that these five centriolar satellite proteins participate in centriole duplication by interacting with different MCPH proteins and delivering them to centrosome. However, they failed to show any insightful mechanisms accounting for the recruitment and function. More specifically, the authors failed to provide direct evidence to support that SPAG5, CEP72, CEP131, MNR, and CEP90 participate in centriole duplication through interaction with MCPH proteins. The binding sites essential for these interactions could be mapped and the binding deficient mutants should be expressed in cells with knockdown background to test whether the interactions are the main reason for the phenotype.

B) Figure 3: since Figure 1 shows that the four MCPH proteins interact with each other, the IPs of CEP72 and SPAG5 should be probed as well for CEP152, WDR62 and CEP63, to test if these two satellite proteins interact exclusively with CDK5RAP2, as suggested, or also with the other 3 MCPH proteins.

C) To support their sequential recruitment model, the authors need to demonstrate that CEP152, WDR62, and Cep63 are not co-depleted after Cep72 or SPAG5 RNAi. If they are, this may argue against the recruitment model and it would just be an issue of total protein level. Similar control experiments apply to the IPs and depletions of the other satellite proteins shown in subsequent figures.

D) It is important to test interaction of the 4 MCPH proteins upon depletion of any of their respective centriolar satellite proteins. This would be required to test whether the recruitment hierarchy occurs at the centrosome, as proposed, or already in the cytoplasm, at the level of a pre-assembled large complex, for example. The authors could repeat the IPs shown in Figure 1 in the absence of SPAG5, CEP72, CEP131, MNR, or CEP90, to test whether the interactions between the 4 MCPH are maintained and, if not, which interactions become disrupted.

2) None of the immunofluorescence data that investigate centrosome localization, and which extends over multiple figures, have been quantified. This is particularly problematic in cases in which the presented data clearly contradicts previously published work. Here it will be crucial to establish that the current data is correct. In addition, quantification of centrosome staining intensities in multiple cells and experiments should be shown after RNAi; this is standard in the field and required to determine how strongly one component depends on another and how variable the phenotype is.

3) Centriole duplication was scored in cells that were in S-phase. That only S phase cells were scored may actually overestimate the phenotype, which could simply be a delay in duplication rather than a real duplication defect.

4) The authors discuss acute vs. prolonged depletion to explain the fact that they see a role for CDK5RAP2 in centriole duplication, which was not observed in previous studies. However, depletion of CDK5RAP2 (also known as Cep215) by RNAi has been used in previous studies, but centriole duplication defects were not been described. Rather CDK5RAP2 seems to have a role in regulating centrosome cohesion and centriole engagement. To support their finding, the authors should perform depletion experiments with siRNA and cells used in previous studies to see whether they can reproduce the duplication defect. This could also be used to confirm the dependency of Cep152 on CDK5RAP2. The same applies to Cep63 depletion. In [5], and [56], CEP63 was shown to be required for CEP152 localization to the centrosomes in different cell types, in conditions of genomic deletion but also RNAi.

5) Figure 6: what is the phenotypical consequence for the cells of expressing the mutant CEP90? Is centriole duplication indeed impaired? I propose that the authors deplete CEP90 and try to rescue the effects of the depletion with the WT CEP90 or with the mutant, analyzing the localization of CEP63 and centriole number.

6) [5], and [56] showed that CEP63 was required for CEP152 localization to the centrosomes in different cell types, upon RNAi knockdown and genomic deletion. In the current manuscript CEP63 depletion does not seem to affect CEP152. The argument in the discussion that acute vs. prolonged treatment may explain discrepancies with previous work is not valid, since acute depletion was also used in some of the previously published experiments. It is important that the authors rigorously address issues regarding previously published, conflicting data. The authors should test siRNA and cells that were previously tested.

7) To confirm the general relevance of their study, and perhaps address inconsistencies with other studies (see #6), the authors should demonstrate that at least some key findings are not only valid in HeLa cells but also in other unrelated cell types.

8) There is human patient data (sequence and phenotype), but it is not clear whether this is IRB approved or exempt.

Other points:

1) In Figure 1—figure supplement 3, the localization/pattern of CDK5RAP2 is entirely different from the one shown in Figure 1. The authors need to provide better images or explain the differences.

2) In Figure 2—figure supplement 1, in the DMSO panel, CDK5RAP2 appears more focused than on Figure 1. This needs to be clarified. In the nocodazole panel, the signal intensity for each of the 4 MCPH proteins appears brighter than in the DMSO panel. Is there indeed more protein co-localizing with PCM1 or is this just due to increased exposure/brightness settings? Here the signal of each protein at the centrioles (centrin marker) versus satellite (PCM1 marker) should be quantified for each condition in a number of cells, to show that there is a de-localization upon nocodazole treatment.

3) In Figure 7—figure supplement 2, there is a reference to a CCDC14 siRNA #3, which doesn't appear before (only #1 and #2). This needs to be clarified. Another issue is that when CCDC14 is depleted, the CEP63 signal is restricted to the centrioles, contrary to what is shown in Figure 7).

4) The single letter/number abbreviations for protein names used as labels for the IF panels are not very intuitive, since letters/numbers from random positions of the protein names are used, it is difficult to guess what “C”, “2” etc. means.

5) Figure 7—figure supplement 5: the channel color legend is missing.

6) In Figure 7, the authors use U2OS cells instead of HeLa (which were used in the previous experiments), without any reference to why the change in cell line.

7) It is a bit confused why CEP135, STIL and SAS4 are not included in this study. Do these MCPH proteins also associate with centriolar satellite partners as proposed in this study?

8) It is intriguing to propose the pairing hypothesis. However, the criteria for the paring proteins identified through MS are not revealed? For example, why CEP250 and CEP170 are not examined in this study? How to exclude the possibility of that these centriolar satellite proteins affect the hemostasis of centriolar satellite rather than the proposed protein-protein interaction?

9) It will be more convincing to include another centrosome protein like CP110 in every panel of IP result as a proper negative control.

[Editors' note: further revisions were requested prior to acceptance, as described below.]

Thank you for resubmitting your work entitled “Centriolar satellites assemble centrosomal microcephaly proteins to recruit CDK2 and promote centriole duplication” for further consideration at *eLife*. Your revised article has been favorably evaluated by Vivek Malhotra (Senior editor), a Reviewing editor, and three reviewers. The manuscript has been improved but there are some remaining issues that need to be addressed before acceptance, as outlined below.

The reviewers of your revised manuscript are satisfied with the experimental aspects, however, one reviewer requests that you address the following item in your Discussion.

This reviewer had raised this issue in the first round of review (reviewer #1 point 1D). The concern is with your proposal of the recruitment model by demonstrating that the sequential interactions between CDK5RAP2, CEP152, WDR62 and CEP63 occur at the centrosome rather than in the cytosol. In this reviewer's opinion the provided immunoprecipitations of MCPH proteins from soluble cell extract in the absence of specific satellite proteins support his/her speculation that the centrosomal recruitment of the MCPH proteins occurs in the form of large complexes that are pre-assembled in the cytosol, in a sequential manner and dependent on the respective satellite protein partner, rather than MCPH proteins being delivered by their satellite partner one by one to the centrosome. The reviewer is concerned with your conclusion that sequential assembly occurs at the centrosome?

The reviewer assumes that the IPs were performed with soluble cell extracts that did not contain centrosomes (because these would end up in the pellet fraction when clearing the extract by high speed centrifugation). Or do you have data indicating the presence of centrosomes in your IPs? If centrosomes are not present during the IPs then the data clearly shows that all 4 MCPH proteins interact in the cytosol, and that the formation of this large complex depends on the sequential recruitment of each MCPH protein by its satellite partner. CDK5RAP2 could be considered the platform onto which all others sequentially and hierarchically assemble. And all of this can occur in the cytosol, in the absence of centrosomes. Obviously this will result in a hierarchical centrosome localization effect, as a result of changes in the pre-assembled cytosolic complexes. Unless you can exclude this possibility, the description and discussion of the recruitment data (and thus the model) needs to be changed.

---

## [Author Response]

This work by Kodani and colleagues from the Reiter lab explored interactions between different microcephaly-associated proteins and between microcephaly-associated proteins and centriolar satellite proteins. The authors seek to make the point that centriolar satellites are crucial for centrosome biogenesis and functionally link satellite proteins to MCPH. The paper is descriptive and functional studies are at the level of dependence of one protein for the localization of another. In spite of the title there is no data showing that satellites bring microcephaly proteins to centrosomes.

We have lengthened the title to more accurately reflect the data shown.

The paper is difficult to read and will be a challenge for anyone not directly involved in this field. Perhaps a model or diagram would help to graphically illustrate the data.

As suggested, we have extensively revised the text to make it easier for the casual reader to interpret, and have added a model to Figure 7 which, together with the schematic demonstrating the requirements for centrosomal localization, summarizes the main points of the manuscript.

Major points:

1) The hierarchical centrosome recruitment model has not been established thoroughly:

A) The authors showed that depletion of centriolar satellite proteins SPAG5, CEP72, CEP131, MNR, and CEP90 disrupted centriole duplication. They claimed that these five centriolar satellite proteins participate in centriole duplication by interacting with different MCPH proteins and delivering them to centrosome. However, they failed to show any insightful mechanisms accounting for the recruitment and function. More specifically, the authors failed to provide direct evidence to support that SPAG5, CEP72, CEP131, MNR, and CEP90 participate in centriole duplication through interaction with MCPH proteins. The binding sites essential for these interactions could be mapped and the binding deficient mutants should be expressed in cells with knockdown background to test whether the interactions are the main reason for the phenotype.

To try to address the request to map the binding sites and perform rescue experiments to support insight into how satellite proteins participate in centriole duplication, we have misexpressed tagged centriolar satellite proteins. Unfortunately, we have found that low levels of expression of centriolar satellite proteins using minimal amounts of DNA leads to the formation of cytoplasmic aggregates, even in knockdown backgrounds. The aggregates started to form after less than 12 hours of expression (Figure 8), insufficient time for cells to go through a centriole duplication cycle, and caused the mislocalization of PCM1, thus making it difficult to interpret the results. Due to the failure to localize as endogenous proteins do, we are hesitant to make any conclusion from these experiments. We are continuing to troubleshoot the expression levels, and we apologize that we were unable to complete this experiment in the two months allotted for revisions.

Author response image 1.**DOI:**
http://dx.doi.org/10.7554/eLife.07519.039

*B)*
Figure 3*: since*
Figure 1
*shows that the four MCPH proteins interact with each other, the IPs of CEP72 and SPAG5 should be probed as well for CEP152, WDR62 and CEP63, to test if these two satellite proteins interact exclusively with CDK5RAP2, as suggested, or also with the other 3 MCPH proteins.*

To further investigate whether CEP72 and SPAG5 interact exclusively with CDK5RAP2, we examined whether they could co-precipitate CEP152, WDR62 and CEP63, as suggested. We determined that CEP72 and SPAG5 specifically interact with CDK5RAP2. The new data are included in Figure 3—figure supplement 1. Text describing these findings is now included in subsection “MCPH proteins interact with centriolar satellite components” of the manuscript.

C) To support their sequential recruitment model, the authors need to demonstrate that CEP152, WDR62, and Cep63 are not co-depleted after Cep72 or SPAG5 RNAi. If they are, this may argue against the recruitment model and it would just be an issue of total protein level. Similar control experiments apply to the IPs and depletions of the other satellite proteins shown in subsequent figures.

We explored the possibility that CEP72 or SPAG5 might be required to stabilize CEP152, WDR62 and CEP63, which would certainly argue against our proposal that they function in CDK5RAP2 recruitment to the centrosome, and only indirectly in the centrosomal recruitment of CEP152, WDR62 and CEP63. As suggested, we checked whether CEP152, WDR62 and CEP63 were co-depleted upon knockdown of CEP72 or SPAG5 and found that they were not. The new data are included in Figure 3—figure supplement 3.

We also examined the stability of CDK5RAP2, CEP152, WDR62 and CEP63 in cells depleted of *CEP131, MNR, CEP90* or *CCDC14* and found levels of the MCPH-associated proteins to be unaltered by depletion of any of these centriolar satellite protein, which is also consistent with the model that they participate in MCPH-associated protein recruitment to the centrosome. The new data are included as Figure 4—figure supplement 2, Figure 5—figure supplement 1, Figure 6—figure supplement 1, and Figure 7—figure supplement 2.

*D) It is important to test interaction of the 4 MCPH proteins upon depletion of any of their respective centriolar satellite proteins. This would be required to test whether the recruitment hierarchy occurs at the centrosome, as proposed, or already in the cytoplasm, at the level of a pre-assembled large complex, for example. The authors could repeat the IPs shown in*
Figure 1
*in the absence of SPAG5, CEP72, CEP131, MNR, or CEP90, to test whether the interactions between the 4 MCPH are maintained and, if not, which interactions become disrupted.*

As suggested, we repeated the immunoprecipitations in the absence of centriolar satellite proteins to examine whether the recruitment hierarchy occurs at the centrosome or as a pre-assembled complex in the cytoplasm. Following depletion of CEP131, MNR or CEP90, immoprecipitation of CEP152, WDR62 or CEP63, respectively, demonstrated that they were no longer able to co-precipitate the other MCPH-associated proteins. For example, CEP152 was unable to bind to CDK5RAP2 or WDR62 in the absence of CEP131. Similarly, WDR62 could not bind CEP152 or CEP63 in MNR-depleted cells and CEP63 failed to bind WDR62 in the absence of CEP90. Together, these data suggest that, in accordance with a possibility raised by the reviewers, the recruitment hierarchy of four MCPH-associated proteins occurs at the centrosome. These new data are included in Figure 4—figure supplement 3, Figure 5—figure supplement 2 and Figure 6—figure supplement 2. Text describing these findings is included in subsections “Centriolar satellite protein CEP131 is required for MCPH-associated protein CEP152 to localize to the centrosome”, “Centriolar satellite protein MOONRAKER is required for MCPH-associated protein WDR62 to localize to the centrosome” and “Centriolar satellite protein CEP90 is required for MCPH-associated protein CEP63 to localize to the centrosome”.

2) None of the immunofluorescence data that investigate centrosome localization, and which extends over multiple figures, have been quantified. This is particularly problematic in cases in which the presented data clearly contradicts previously published work. Here it will be crucial to establish that the current data is correct. In addition, quantification of centrosome staining intensities in multiple cells and experiments should be shown after RNAi; this is standard in the field and required to determine how strongly one component depends on another and how variable the phenotype is.

As advised, we quantified the fluorescence intensities of the immunostaining of MCPH-associated proteins at centrosomes in cells depleted of MCPH-associated proteins or their cognate centriolar satellite protein. The data is included as Figure 1—figure supplement 3 and Figure 1—figure supplement 7, 8D; Figure 3—figure supplement 3, Figure 4—figure supplement 2; Figure 5—figure supplement 1; Figure 6—figure supplement 1; and Figure 7—figure supplement 2.

3) Centriole duplication was scored in cells that were in S-phase. That only S phase cells were scored may actually overestimate the phenotype, which could simply be a delay in duplication rather than a real duplication defect.

To investigate the possibility that the duplication phenotype is due to a delay in duplication rather than a real centriolar duplication defect, we analyzed cells in mitosis treated with siRNA to SC, CDK5RAP2, CEP152, WDR62 or CEP63. We found that the duplication defect described in the text is also present during mitosis, suggesting that the phenotype is due to a duplication defect that is not attributable to a delay in duplication. The new data are included in Figure 1—figure supplement 4 and described in the subsection “MCPH-associated proteins interact and hierarchically localize to the centrosome”.

*4) The authors discuss acute vs. prolonged depletion to explain the fact that they see a role for CDK5RAP2 in centriole duplication, which was not observed in previous studies. However, depletion of CDK5RAP2 (also known as Cep215) by RNAi has been used in previous studies, but centriole duplication defects were not been described. Rather CDK5RAP2 seems to have a role in regulating centrosome cohesion and centriole engagement. To support their finding, the authors should perform depletion experiments with siRNA and cells used in previous studies to see whether they can reproduce the duplication defect. This could also be used to confirm the dependency of Cep152 on CDK5RAP2. The same applies to Cep63 depletion. In*
[5]*, and*
[56]*, CEP63 was shown to be required for CEP152 localization to the centrosomes in different cell types, in conditions of genomic deletion but also RNAi.*

We tested whether CEP215/CDK5RAP2 depletion using the siRNA previously described by Graser et al. altered centriole duplication. Cells treated with either the siRNA they call *CEP215 #282* or *#283* did caused a failure of centriolar duplication, and a failure to recruit CEP152, WDR62 or CEP63 to the centrosome, in agreement with our findings using independent siRNAs (CDK5RAP2 #1 and 2). Interestingly, siRNA *CEP215 #283* caused a centriole disengagement phenotype, but we did not observe an alteration in centriole engagement in cells treated with siRNA *CEP215 #282* or either of our siRNAs (CDK5RAP2 #1 and 2) treated cells did not disrupt centriole engagement. We have included the new data as a Figure 1—figure supplement 3 and described the results in the subsection “MCPH-associated proteins interact and hierarchically localize to the centrosome”. As for the question about the relationship of CEP63 and CEP152, please refer to the answer to point 6, below.

*5)*
Figure 6*: what is the phenotypical consequence for the cells of expressing the mutant CEP90? Is centriole duplication indeed impaired? I propose that the authors deplete CEP90 and try to rescue the effects of the depletion with the WT CEP90 or with the mutant, analyzing the localization of CEP63 and centriole number.*

We agree with the reviewer that these are interesting questions. Unfortunately, even misexpression of wild-type or mutant CEP90 with a weak promoter (EF1alpha) for less than a cell cycle leads to the formation of large cytoplasmic aggregates of CEP90. Given that these cytoplasmic aggregates may have cellular consequences independent of the normal functions of CEP90, we were unable to conclude whether centriole duplication is rescued specifically if the protein of interest is not localized properly. Given that the human *CEP90* mutation is not central to the main points of the paper, we are happy to remove these data and publish them in a subsequent work in which the phenotypical consequences can be more fully elucidated if that is the opinion of the reviewers. Please refer to the answer to point 1, in which we show cells misexpressing WT and E89Q mutant CEP90 that leads to the formation of cytoplasmic aggregates that misrecruit the centriolar satellite, PCM1.

*6)*
[5]*, and*
[56]
*showed that CEP63 was required for CEP152 localization to the centrosomes in different cell types, upon RNAi knockdown and genomic deletion. In the current manuscript CEP63 depletion does not seem to affect CEP152. The argument in the discussion that acute vs. prolonged treatment may explain discrepancies with previous work is not valid, since acute depletion was also used in some of the previously published experiments. It is important that the authors rigorously address issues regarding previously published, conflicting data. The authors should test siRNA and cells that were previously tested.*

We investigated whether our hierarchy model held true for CEP63 using siRNAs reported by Brown et al. While we would have liked to confirm the experiments from Sir et al., Dhamacon has discontinued the SMARTpool siRNA product referred to in their Nature Genetics paper, and was unable to tell us the siRNA sequences. We found that depletion of CEP63 using the siRNA that they call *CEP63 NB* disrupted centriole duplication, centrosomes failed to accumulate CDK5RAP2, CEP152, and WDR62, in accordance with what they reported and in contrast to our findings using other ways of depleting CEP63. Examining protein levels revealed that the *CEP63 NB* siRNA depleted endogenous CEP63 protein levels and also led to the co-depletion of CDK5RAP2, CEP152, and WDR62, again in contrast to our findings using other ways of depleting CEP63. The independent siRNAs *CEP63 #1* and *#2* deplete endogenous CEP63 to an equivalent degree but do not cause the co-depletion of these other proteins*.* Throughout this work, we have been careful to report only phenotypes that are caused by siRNAs that both efficiently deplete the target protein, and show phenotypes that are caused by multiple independent siRNAs. While our model would predict that destabilization of CDK5RAP2, as occurs upon knockdown with *CEP63 NB*, will cause a failure to recruit CEP152 to the centrosome, we are hesitant to make this conclusion as it is only observed with the use of this siRNA. These data are included as a Figure 1–figure supplement 8 and described the results in the subsection “MCPH-associated proteins interact and hierarchically localize to the centrosome”.

7) To confirm the general relevance of their study, and perhaps address inconsistencies with other studies (see #6), the authors should demonstrate that at least some key findings are not only valid in HeLa cells but also in other unrelated cell types.

As suggested, we tested key findings using an unrelated cell type. We confirmed that CDK5RAP2, CEP72 and SPAG5 are required for centriole duplication in U2OS cells. We also confirmed that the protein stability of CDK5RAP2 depends CEP72 and SPAG5 in U2OS cells. We were also able to confirm that the centrosomal localization of CDK5RAP2 depends on CEP72 and SPAG5 where it serves as a scaffold to recruit CEP152, WDR62, and CEP63. We have included these new data as Figure 3—figure supplement 2 and included this in the subsection “Centriolar satellite proteins CEP72 and SPAG5 are required for MCPH-associated protein CDK5RAP2 to localize to the centrosome”.

8) There is human patient data (sequence and phenotype), but it is not clear whether this is IRB approved or exempt.

The human patient data is IRB approved and this approval is described in the Materials and methods section.

Other points:

*1) In*
Figure 1—figure supplement 3*, the localization/pattern of CDK5RAP2 is entirely different from the one shown in*
Figure 1*. The authors need to provide better images or explain the differences.*

The appearance of CDK5RAP2 localization does differ using diffraction limited microscopy (Figure 1) versus structured illumination microscopy (SIM, Figure 1—figure supplement 3). The difference is attributable to the properties of SIM. The low contrast staining of CDK5RAP2 outside of the toroidal structure at the center of the centrosome creates a situation for SIM very similar to the in-focus image surrounded by out-of-focus light. In the resulting point spread function, the data from the neighboring fluorophores outnumbers the data from the central fluorophore and the SIM algorithm’s high-pass filters cannot assign the data from these fluorophores in reciprocal space to coordinates based on where the light originated from. Thus, low contrast images are not distinguishable from noise even with the use of different Wiener filter values in SIM. This is a known limitation of the use of SIM with low contrast samples. SIM is more appropriate use with high contrast, structured images. Thus, we have limited our use of SIM to imaging the region of CDK5RAP2 at the toroidal structure which, unlike the surrounding CDK5RAP2 domain, is both high contrast and highly structured. We have included discussion of this in subsection “MCPH-associated proteins interact and hierarchically localize to the centrosome”.

*2) In*
Figure 2—figure supplement 1*, in the DMSO panel, CDK5RAP2 appears more focused than on*
Figure 1*. This needs to be clarified.*

We thank the reviewer for bringing this to our attention. We have replaced the panel with a more representative image.

In the nocodazole panel, the signal intensity for each of the 4 MCPH proteins appears brighter than in the DMSO panel. Is there indeed more protein co-localizing with PCM1 or is this just due to increased exposure/brightness settings? Here the signal of each protein at the centrioles (centrin marker) versus satellite (PCM1 marker) should be quantified for each condition in a number of cells, to show that there is a de-localization upon nocodazole treatment.

We concur that the signal intensity of each of the four MCPH-associated proteins appears brighter upon nocodazole treatment. We were careful to image the DMSO and nocodazole-treated samples at the same exposure settings and process all images in parallel and in the same way. As suggested by the reviewers, we quantified the fluorescence signal of each MCPH-associated protein at the centrioles and satellites and found that nocodazole increased the proportion of the MCPH-associated proteins at the satellites. This quantitation is now included as a graph in Figure 2—figure supplement 1.

*3) In*
Figure 7—figure supplement 2*, there is a reference to a CCDC14 siRNA #3, which doesn't appear before (only #1 and #2). This needs to be clarified. Another issue is that when CCDC14 is depleted, the CEP63 signal is restricted to the centrioles, contrary to what is shown in*
Figure 7*).*

We thank the reviewers for bringing this mistake to our attention. We have changed the reference to siRNA #3 to siRNA #2. We apologize for the mistake. Although the domain of CEP63 localization expands with the increase in Centrin foci upon CCDC14 depletion, we do not perceive the CEP63 to be strictly localized to centrioles. To further assess the accumulation of CEP63 to Centrin positive foci in *CCDC14*-depleted cells, we quantified the fluorescence intensity of CEP63 and the other three MCPH-associated proteins and found that only CEP63 co-localized with the majority of the Centrin positive foci. This quantitation is included in Figure 7—figure supplement 2.

4) The single letter/number abbreviations for protein names used as labels for the IF panels are not very intuitive, since letters/numbers from random positions of the protein names are used, it is difficult to guess what “C”, “2” etc. means.

We apologize for not using single letter/number abbreviations that were not intuitive. To rectify this, we replaced all labels with their protein names. Additionally, the color of the font used for label of each protein either on the top or left of each panel reflects the channel used to image that protein, making it clear that Centrin is always depicted in green and the other centrosomal protein is depicted in red.

*5)*
Figure 7—figure supplement 5*: the channel color legend is missing.*

We thank the reviewer for bringing this to our attention. We have added the channel color legend to the figure.

*6) In*
Figure 7*, the authors use U2OS cells instead of HeLa (which were used in the previous experiments), without any reference to why the change in cell line.*

While both U2OS and HeLa cells displayed supernumerary Centrin foci upon CCDC14 depletion, supernumerary Centrin foci of HeLa cells are tightly packed, making them difficult resolve and assess how they co-localize with other centrosomal proteins. Therefore, we displayed U2OS cells, in which the supernumerary Centrin foci are more dispersed. We made note of this rationale in subsection “Centriolar satellite component CCDC14 restrains the centrosomal accumulation of CEP63 and CDK2”.

7) It is a bit confused why CEP135, STIL and SAS4 are not included in this study. Do these MCPH proteins also associate with centriolar satellite partners as proposed in this study?

We agree with the reviewer that CEP135, STIL and SAS4 would have been nice to include in this study. However, because we do not have reagents of sufficient quality to fully assess whether CEP135, STIL and SAS4 associate with still other centriolar satellite proteins, we will need to work on developing better tools. It is possible that CEP135, STIL and SAS4 bind cognate centriolar satellite proteins similar to CDK5RAP2, CEP152, WDR62 and CEP63, but for this work, we focused on the four MCPH-associated proteins that we were confident biochemically interacted with each other, and the six satellite proteins that we were confident interacted with them.

8) It is intriguing to propose the pairing hypothesis. However, the criteria for the paring proteins identified through MS are not revealed? For example, why CEP250 and CEP170 are not examined in this study? How to exclude the possibility of that these centriolar satellite proteins affect the hemostasis of centriolar satellite rather than the proposed protein-protein interaction?

The criteria for our focusing on the proteins we identified were twofold: we focused on proteins that both localized to the centrosome and which were previously implicated in centriole duplication. While both CEP250 and CEP170 localize to the centrosome, neither was strongly required for centriole duplication in our hands. Whereas CEP250 and CEP170 have roles in centriole cohesion and subdistal appendage organization, respectively, we were not able to discern any roles for them in MCPH-associated protein function, and thus were not included in this study. Inhibition of the function of centriolar satellite proteins may well be affecting aspects of centrosomal biology beyond the interaction with MCPH-associated proteins, including homeostasis of the centriolar satellites. However, we have focused on the aspects of centrosomal function for which we have positive evidence, and will continue to assess whether these centriolar satellite proteins participate in other aspects of centrosomal function in the future. A more complete description of these possibilities is now included in the Discusssion.

9) It will be more convincing to include another centrosome protein like CP110 in every panel of IP result as a proper negative control.

We thank the reviewers for suggesting these sets of experiments. As an extra measure of specificity, we now include CP110 as a negative control in Figures 1 and 2.

[Editors' note: further revisions were requested prior to acceptance, as described below.]

The reviewers of your revised manuscript are satisfied with the experimental aspects, however, one reviewer requests that you address the following item in your Discussion.

This reviewer had raised this issue in the first round of review (reviewer #1 point 1D). The concern is with your proposal of the recruitment model by demonstrating that the sequential interactions between CDK5RAP2, CEP152, WDR62 and CEP63 occur at the centrosome rather than in the cytosol. In this reviewer's opinion the provided immunoprecipitations of MCPH proteins from soluble cell extract in the absence of specific satellite proteins support his/her speculation that the centrosomal recruitment of the MCPH proteins occurs in the form of large complexes that are pre-assembled in the cytosol, in a sequential manner and dependent on the respective satellite protein partner, rather than MCPH proteins being delivered by their satellite partner one by one to the centrosome. The reviewer is concerned with your conclusion that sequential assembly occurs at the centrosome?

The reviewer assumes that the IPs were performed with soluble cell extracts that did not contain centrosomes (because these would end up in the pellet fraction when clearing the extract by high speed centrifugation). Or do you have data indicating the presence of centrosomes in your IPs? If centrosomes are not present during the IPs then the data clearly shows that all 4 MCPH proteins interact in the cytosol, and that the formation of this large complex depends on the sequential recruitment of each MCPH protein by its satellite partner. CDK5RAP2 could be considered the platform onto which all others sequentially and hierarchically assemble. And all of this can occur in the cytosol, in the absence of centrosomes. Obviously this will result in a hierarchical centrosome localization effect, as a result of changes in the pre-assembled cytosolic complexes. Unless you can exclude this possibility, the description and discussion of the recruitment data (and thus the model) needs to be changed.

Our data show that each MCPH protein cannot interact with other MCPH proteins in the absence of its cognate centriolar satellite interactor, and that there is a sequential MCPH protein recruitment to the centrosome.

Our immunoprecipitation experiments were performed with soluble cell extracts that likely contain the majority of centrosomal proteins aside from the salt-insoluble centriole core (Schnackenberg 1999, Flory 2000, Schnackenberg 1998). As most non-centriolar components are extracted off of the centrosome by the combination of salt and detergent that we use, we surmise that the MCPH components are part of the soluble cell extract used in our IPs. Thus, while centrosomes are not present in our lysates, many centrosomal proteins are.

We concur with the reviewer that, based on the available information, it is not possible to discriminate between formation of the MCPH protein complex at the centrosome and pre-assembly into a complex in the cytoplasm prior to being delivered to the centrosome. We agree that it is possible that the failure to assemble the MCPH complex in the cytoplasm could result in the failed hierarchical assembly at the centrosome. Because we cannot definitively distinguish between assembly of the MCPH complex in the cytoplasm and at the centrosome, we have refined the Discussion and the legend for the final model figure to specifically discuss both possibilities.